**Investigation**

# Distinct genomic contexts predict gene presence–absence variation in different pathotypes of *Magnaporthe oryzae*

Pierre M. Joubert,[1,2,*] Ksenia V. Krasileva [1,2,*]

[1]Department of Plant and Microbial Biology, University of California-Berkeley, Berkeley, CA 94720, USA
[2]Center for Computational Biology, University of California-Berkeley, Berkeley, CA 94720, USA

*Corresponding author: Department of Plant and Microbial Biology, University of California-Berkeley, Koshland Hall, Berkeley, CA 94720, USA. Email: pierrej@berkeley.edu;
*Corresponding author: Department of Plant and Microbial Biology, University of California-Berkeley, Koshland Hall, Berkeley, CA 94720, USA. Email: kseniak@berkeley.edu

Fungi use the accessory gene content of their pangenomes to adapt to their environments. While gene presence–absence variation contributes to shaping accessory gene reservoirs, the genomic contexts that shape these events remain unclear. Since pangenome studies are typically species-wide and do not analyze different populations separately, it is yet to be uncovered whether presence–absence variation patterns and mechanisms are consistent across populations. Fungal plant pathogens are useful models for studying presence–absence variation because they rely on it to adapt to their hosts, and members of a species often infect distinct hosts. We analyzed gene presence–absence variation in the blast fungus, *Magnaporthe oryzae* (syn. *Pyricularia oryzae*), and found that presence–absence variation genes involved in host–pathogen and microbe–microbe interactions may drive the adaptation of the fungus to its environment. We then analyzed genomic and epigenomic features of presence–absence variation and observed that proximity to transposable elements, gene GC content, gene length, expression level in the host, and histone H3K27me3 marks were different between presence–absence variation genes and conserved genes. We used these features to construct a model that was able to predict whether a gene is likely to experience presence–absence variation with high precision (86.06%) and recall (92.88%) in *M. oryzae*. Finally, we found that presence–absence variation genes in the rice and wheat pathotypes of *M. oryzae* differed in their number and their genomic context. Our results suggest that genomic and epigenomic features of gene presence–absence variation can be used to better understand and predict fungal pangenome evolution. We also show that substantial intra-species variation can exist in these features.

Keywords: plant pathogen; fungi; presence–absence variation; evolution; *Magnaporthe oryzae*; *Pyricularia oryzae*; structural variation; population genetics; comparative genomics; machine learning

## Introduction

Many microbial species have expansive pangenomes that allow them to adapt to their environments. While bacteria typically gain and lose genes in the form of large horizontal gene transfer events (McInerney *et al.* 2017), the accessory portions of fungal pangenomes, tend to be shaped by point mutations, small DNA insertions and deletions (indels) which cause changes in gene sequences, and large indels which cause insertions or deletions of entire gene sequences, all of which contribute to gene presence–absence variation (PAV) (Martin 2017; McCarthy and Fitzpatrick 2019). Many previous fungal pangenome studies have analyzed the differences between core and accessory genes and found that accessory genes tend to be enriched in functions important for rapid adaptation (McCarthy and Fitzpatrick 2019; Badet *et al.* 2020; Kaushik *et al.* 2022; Moolhuijzen *et al.* 2022). Studying the features and genomic contexts in which PAV occurs in fungal genomes could therefore help us better understand and predict rapid evolution in these organisms. In pursuit of this understanding, some studies have characterized an association of accessory genes with chromosome ends (subterminal regions) and transposable elements (TEs) (McCarthy and Fitzpatrick 2019; Badet *et al.* 2020). One recent study constructed models that could predict

meiotically derived structural variation generation events in *Zymoseptoria tritici*, and identified TEs, histone marks and GC content as particularly important predictors (Badet *et al.* 2021). This research raises the exciting possibility that these models could be used to predict PAV events in other fungi and could help us better understand fungal pangenome evolution. However, whether different mechanisms that generate gene PAV are associated with distinct genomic contexts remains an active area of research. Furthermore, it has remained unclear whether any patterns in genomic or epigenomic features of PAV events could be generalized to all populations of the same species, as pangenomes are typically assembled for entire species without consideration of differential evolution between populations.

Fungal plant pathogens are useful models for studying pangenome evolution. They have dynamic pangenomes that allow them to adapt to their hosts and secrete a wide range of rapidly evolving effector proteins to cause disease (Badet *et al.* 2020; Kaushik *et al.* 2022; Moolhuijzen *et al.* 2022). These effectors can become a disadvantage, however, when the immune receptors of their hosts acquire new recognition specificities which allow the receptors to detect these effectors and trigger an immune response (Tamborski and Krasileva 2020). Gene PAV plays an important

role in avoiding this response (Sánchez-Vallet *et al.* 2018). Point mutations in effectors also contribute to avoiding detection by the host. Therefore, effectors tend to be located in TE-dense and gene-poor regions of the genome, which helps effectors rapidly evolve and escape recognition, while slower evolution and house-keeping genes occur in TE-poor and gene-dense regions of the genome (Dong *et al.* 2015; Torres *et al.* 2020). This idea is often referred to as the "two-speed" genome concept. Effectors tend to be prone to PAV but it is currently unclear whether the two-speed genome concept applies directly to gene PAV (Sánchez-Vallet *et al.* 2018). Finally, many fungal plant pathogen species are made up of populations that infect distinct hosts. The existence of these pathotypes makes fungal plant pathogens useful models for characterizing and comparing gene PAV across different populations of the same species and investigating how this PAV might affect differences in their phenotypes.

*Magnaporthe oryzae* (syn. *Pyricularia oryzae*) causes the blast disease of many grasses, including rice and wheat, and is amongst the most important and well-studied pathogens with hundreds of available genomes and next-generation sequencing datasets (Dean *et al.* 2012; Ceresini *et al.* 2019). The fungus has been reported to experience substantial gene PAV but these analyses have been largely restricted to effectors, and the genomic and epigenomic features associated with these PAV events remain largely unexplored (Kim *et al.* 2019; Latorre *et al.* 2020; Thierry *et al.* 2022). Since *M. oryzae* is thought to be mostly clonal, the study of how its pangenome can evolve without substantial recombination is also possible within this species (Gladieux, Ravel, *et al.* 2018b; Thierry *et al.* 2022; Rahnama *et al.* 2023). The *M. oryzae* species is made up of many pathotypes and host-specificity is largely monophyletic (Gladieux, Condon, *et al.* 2018). Exchange of genetic information between host-specific isolates is thought to be rare, though there is clear evidence of such exchanges (Gladieux, Condon, *et al.* 2018; Rahnama *et al.* 2023). The isolation of the different host-specific populations therefore enables the comparison of gene PAV between pathotypes within the *M. oryzae* species. While rice blast has been a long-standing threat, the rapid spread of wheat blast throughout the world as well as its particularly devastating effect on wheat crops has fueled research into the wheat-infecting pathotype of *M. oryzae* that causes this disease and especially how it was able to jump hosts from rice to wheat and become such a devastating pathogen (Ceresini *et al.* 2019). Therefore, it is especially important to investigate whether differences exist in gene PAV between these 2 pathotypes. Altogether, *M. oryzae* offers a unique opportunity to study gene PAV and the genomic and epigenomic features that shape these events as well as how these events vary within a species.

In this study, we sought to characterize and compare gene PAV in rice-infecting (MoO) and wheat-infecting *M. oryzae* (MoT). We first identified orthogroups experiencing PAV caused by large indels that distinguished isolated MoO lineages and found that they were enriched in functions related to host–pathogen and microbe–microbe interactions. Next, we characterized the genomic contexts in which these PAV events occur in MoO and MoT and found that TEs were often found in proximity to these genes. Additionally, we found that PAV genes were smaller, had a lower GC content, were less expressed and were more likely to show H3K27me3 histone mark sequencing signal than conserved genes. We used these features to construct a random forest classifier and found that the differences we observed were strong enough to produce a model that predicted whether a gene is likely to experience PAV with high precision (86.06%) and recall (92.88%). Finally, we found significant differences in the number of PAV events and

the features that predict PAV in MoO and MoT, which could reflect their differing evolutionary history and could be evidence of distinct mechanisms contributing to PAV in the 2 recently diverged lineages.

## Methods

### Genome annotation, proteome orthogrouping, and phylogeny generation

The set of 123 MoO genomes was obtained from a previously published study (Zhong *et al.* 2018; Pordel *et al.* 2020; Thierry *et al.* 2022), while 36 MoT genomes as well as a single *Magnaporthe grisea* proteome (GCA004355905.1) were obtained from NCBI's GenBank (Supplementary Table 1). All genomes were verified to have more than 90% completeness using BUSCO version 5.2.2 and the "sordariomycetes_odb10" option (Simão *et al.* 2015). Genomes were annotated using FunGAP (Min *et al.* 2017) version 1.1.0 and RNAseq data was obtained from Sequence Read Archive (SRA) accession ERR5875670. The "sordariomycetes_odb10" option was used for the busco_dataset argument and the "magnaporthe_grisea" option was used for the augustus_species argument. For repeat masking, a TE library generated by combining the RepBase (Bao *et al.* 2015) fngrep version 25.10 with a de novo repeat library, generated by RepeatModeler (Flynn *et al.* 2020) version 2.0.1 run on the *M. oryzae* Guy11 genome (GCA002368485.1) with the LTRStruct option, was used for all genomes. Annotated proteomes were then used as input for OrthoFinder (Emms and Kelly 2019) version 2.5.4 to form 2 separate sets of orthogroups, 1 for MoO proteomes and 1 for MoT proteomes. The *M. grisea* proteome was included in both as an outgroup. Orthogrouping was performed using the "diamond_ultra_sens" parameter for sequence search, the "mafft" parameter for species tree generation and the "fasttree" parameter for gene tree generation. Single copy orthologs (SCOs) were then obtained from the OrthoFinder output, aligned using mafft (Katoh and Standley 2013) version 7.487 with the –maxiterate 1000 parameter and the –globalpair parameter, concatenated, and then trimmed using trimal (Capella-Gutiérrez *et al.* 2009) version 1.4.rev22 and a 0.8 gap threshold parameter. Finally, fasttree (Price *et al.* 2010) version 2.1.10 with the gamma parameter was used to generate a phylogeny and ape (Paradis and Schliep 2019) version 5.5 was used to root each tree on the *M. grisea* outgroup.

### Gene absence validation

A preliminary set of missing orthogroups in each genome was obtained from the OrthoFinder outputs. Gene absences were validated by first using TBLASTN (Camacho *et al.* 2009) version 2.7.1+ with the -max_intron_length 3000 parameter to align all protein sequences from an orthogroup to the genome that was missing that orthogroup. Any orthogroup that resulted in 2 or more alignments with greater than 55% sequence identity, greater than 55% query coverage and an *e*-value smaller than $10^{-10}$ when aligned to the target genome were selected for further verification. TBLASTN hits were converted to GFF format using custom scripts (see *Data availability*) and then converted from GFF format to protein sequences using agat_sp_extract_sequences.pl version 0.9.1 from the AGAT toolkit (https://github.com/NBISweden/AGAT) with default settings, and aligned against all protein sequences in all orthogroups using BLASTP (Camacho *et al.* 2009) version 2.7.1+. Finally, the BLASTP alignments were ranked by *e*-value and the top 100 alignments were collected, as well as which orthogroup the matching protein sequence belonged to. The most common orthogroup within these 100 alignments was used to determine which orthogroup the sequence would have belonged to if it had

been annotated by FunGAP. If no TBLASTN hits were found or if the BLASTP hits did not match the original missing orthogroup, the absence was counted as a validated absence, otherwise it was removed from the set of missing orthogroups.

The filters and cutoffs used for this method were optimized by first re-running orthogrouping as previously described, without the proteome of the MoO isolate CH0043 (chosen at random). A list of orthogroups that were found in the CH0043 proteome in our full dataset orthogrouping was then used as test cases. Our pipeline was optimized by testing different cutoffs for sequence identity, query coverage and e-value, so that these cutoffs were as stringent as possible while still only incorrectly classifying less than 1% of these orthogroups as absent.

## Effector annotation

Effectors were predicted in all proteomes by first selecting genes with signal peptides which were predicted using SignalP (Petersen et al. 2011) version 4.1 using the "euk" organism type and using 0.34 as a D-cutoff for both noTM and TM networks. Genes with predicted transmembrane domains from TMHMM (Krogh et al. 2001) version 2.0c were then excluded. Finally, EffectorP (Sperschneider and Dodds 2022) version 3.0 was used to predict effectors from this secreted gene set. Effector orthogroups were then called if at least half of the orthologs within the orthogroup were annotated as predicted effectors.

## Principal component analysis and identification of lineage-differentiating PAV orthogroups

The matrix of missing effector orthogroups for each MoO isolate was used for principal component analysis (PCA) using the prcomp function in R version 3.6.1. PCA was performed a second time using the matrix of all missing orthogroups. The get_pca_var function in R version 3.6.1 was used to obtain the amount that each orthogroup contributed to the variance of PCs 1 and 2. Orthogroups that contributed more than 0.1% of this variance in either PC1 or PC2 were labeled lineage-differentiating PAV orthogroups. In total, these orthogroups explained 70.53% of the variance in PC1 and 62.17% of the variance in PC2.

## Gene ontology and protein family enrichment analyses

All proteins were annotated for GO terms using the PANNZER2 (Törönen et al. 2018) webserver and command line software SANSPANZ version 3 in October 2022. Only annotations with a positive predictive value greater than 0.6 and an ARGOT rank of 1 were kept. All GO terms assigned to genes within an orthogroup were then transferred to their orthogroup. GO term enrichment analysis was then performed using TopGO (Alexa and Rahnenfuhrer 2023) version 2.36.0 and enrichment was calculated using the Fisher's exact test and the "weight" algorithm. Only GO terms that were assigned to 3 or more lineage-differentiating PAV orthogroups and with enrichment P-values less than 0.05 were reported.

PFAM enrichment analysis was performed by annotating PFAM domains using pfam_scan.pl (Madeira et al. 2022) version 1.6–4 and the PFAM-A database. The output from pfam_scan.pl was parsed using K-parse_Pfam_domains_v3.1.pl (https://github.com/krasileva-group/plant_rgenes) (Sarris et al. 2016) and an e-value cutoff of 0.001, and domain names were simplified by removing numbers and additional letters attached to domain names. Orthogroups were called as containing a domain if at least half of their orthologs had that domain annotation. Fisher's exact test for enrichment was performed using the scipy.stats Python module (Virtanen et al. 2020) version 1.9.0. Only domains which

were observed in 3 or more lineage-differentiating PAV orthologs and with enrichment P-values less than 0.05 were reported.

## Identification of large indels

Illumina sequencing data were obtained from 117 datasets for MoO and 47 datasets for MoT from the SRA (Supplementary Files 1 and 2). Reads were mapped to the M. oryzae Guy11 genome (GCA002368485.1) for MoO datasets and to the M. oryzae B71 genome (GCA004785725.2) for MoT datasets using BWA MEM (Li 2013) version 0.7.17-r1188. Read duplicates were marked using Picard (https://broadinstitute.github.io/picard/) version 2.9.0. Structural variants were then called using smoove (https://github.com/brentp/smoove) version 0.2.8, wham (Kronenberg et al. 2015) version 1.7.0-311-g4e8c, Delly (Rausch et al. 2012) version 0.9.1, and Manta (Chen et al. 2016) version 1.6.0 using default settings. The Delly output was processed using bcftools (Danecek et al. 2021) version 1.6 to keep only called structural variants that passed Delly's quality control. Structural variants were then merged and filtered using SURVIVOR (Jeffares et al. 2017) version 1.0.7. Structural variants that were the same type, were on the same strand, and had breakpoints within 1,000 bp were merged. Only structural variants that were called by 3 or more callers and were larger than 50 bp were kept. Finally, the structural variants called for each dataset were all merged as before except breakpoints within 100 bp were merged. From this list of all structural variants, only structural variants labeled as deletions were kept for further analysis.

## Definition of PAV orthogroups and conserved groups

For each lineage, PAV orthogroups were defined by first taking the matrix of validated PAVs and filtering this matrix to orthogroups that were present in at least 2 isolates and absent in at least 2 isolates. The SCO phylogeny of the lineage was then analyzed for each candidate PAV orthogroup. If the orthogroup was only absent in strains that formed a monophyletic group, the orthogroup was not considered to be a PAV orthogroup. Additionally, if the orthogroup was only found in strains that formed a monophyletic group, the orthogroup was not considered to be a PAV orthogroup either. All orthogroups that were therefore present in 2 independent groups and absent in 2 independent groups were labeled PAV orthogroups. All orthogroups that were missing in 1 or fewer strains were considered conserved orthogroups. All other orthogroups were considered "other".

## Transposable element annotation

TE annotation was performed using RepeatMasker (Smit et al. 2013–2015) version 4.1.1 and a reference TE library for all pathotypes of M. oryzae generated by Nakamoto et al. (2023). The parameters -cutoff 250, -nolow, -no_is, and -norna were used for the RepeatMasker command.

## Next-generation sequencing data and GC content analysis

RNAseq data for MoO were obtained from SRA (Supplementary File 3) from a previously published study (Zhang et al. 2021) and mapped to the M. oryzae Guy11 genome (GCA002368485.1) for in culture data and the M. oryzae Guy11 genome combined with the Oryza sativa Nipponbare genome (GCA001433935.1) for the in planta data. RNAseq data for MoT were obtained from SRA accessions SRR9127598 through SRR9127602 from a previously published study (Peng et al. 2019) and mapped to the M. oryzae B71 genome (GCA004785725.2) for in culture data and the M. oryzae B71 genome

combined with the *Triticum aestivum* genome (GCA900519105.1) for the in planta data. Mapping was performed using STAR (Dobin *et al.* 2013) version 2.7.1a and index files for mapping were made using the previously mentioned genomes and genome combinations along with corresponding gene annotation files obtained from FunGAP for the *M. oryzae* genomes and from GenBank for the rice and wheat genomes or using BED format files of indels identified in this study. Read counts for each gene or indel were calculated using the –quantMode GeneCounts parameter in STAR. These read counts were normalized to gene or indel size as reads per kilobase values (RPK), then the total number of RPKs were summed for each sample and divided by 1 million. This sum was used to normalize read counts in each sample to obtain transcript per million (TPM) values for each sample. These TPM values were then averaged across replicates.

Published ChIP-Seq data for H3K27me3, H3K27ac, and H3K36me3 histone marks were obtained from a study published by Zhang *et al.* (2021). Published eccDNA sequencing data were obtained from a previous study by Joubert and Krasileva (2022). Reads were mapped to the *M. oryzae* Guy11 genome using BWA MEM (Li 2013) version 0.7.17-r1188. Read counts per gene or per indel were obtained using the coverage command from the BEDtools suite of tools (Quinlan and Hall 2010) version 2.28.0. Read counts were normalized for gene or indel size and library size and averaged across replicates as for RNAseq data.

Methylation data from *M. oryzae* mycelium were obtained from a previous study published by Jeon *et al.* (2015). Reads were mapped to the *M. oryzae* genome and processed using the Bismark pipeline (Krueger and Andrews 2011) version 0.24.0. Methylation percentage for all cytosines was extracted while ignoring the first 2 bases of all reads. The percentage of methylated cytosines was then calculated for a gene or indel by averaging the methylation percentage of all cytosines in that gene or indel.

To assign signals from next-generation sequencing datasets to orthogroups, signals for all orthologs in *M. oryzae* Guy11 and *M. oryzae* B71 within each orthogroup were averaged. Any orthogroups that did not have orthologs from B71 and Guy11 within them were given a value equal to the median value for all other orthogroups. *M. oryzae* Guy11 was not included in the original orthogrouping so a separate set of orthogroups were generated which included the *M. oryzae* Guy11 proteome annotated using FunGAP (Min *et al.* 2017) as previously described in order to transfer the next-generation sequencing data signals.

Finally, GC content values for genes, flanking regions, and indels were calculated using the nuc command in BEDTools (Quinlan and Hall 2010) version 2.28.0.

## Window-based density plots of gene and TE content for genomic regions

10 bp windows were first generated for each *M. oryzae* reference genome. The number of TEs and the number of genes in each window were then calculated using the coverage command in BEDTools (Quinlan and Hall 2010) version 2.28.0 and stored as bedgraph files. Bigwig files were generated from bedgraph files using the bedGraphToBigWig tool (https://www.encodeproject.org/software/bedgraphtobigwig/) version 4. Finally, data for window-based density plots of indels were generated using the computeMatrix scale-regions and the plotProfile commands of the DeepTools suite of tools (Ramírez *et al.* 2016) version 3.5.1. Briefly, this tool scales all deletion regions to a set length (in this case, 1kbp) by stretching or compressing them and calculates the number of elements (TEs or genes) in each 10 bp window of these scaled regions. The counts per window are then averaged across all elements.

This process is also done for the flanking regions (in this case, 5kbp) except no scaling is done for these regions.

## Random forest classification and feature importance calculation

Random forest classifiers were trained and performance statistics were calculated using the scikit-learn Python module (Pedregosa *et al.* 2011) version 1.1.1. The hyperparameters used to train the model were as follows: 2,000 estimators, a minimum of 2 samples to split a node, no minimum number of samples per leaf, no maximum tree depth, no maximum number of features per tree, and bootstrapping enabled. Classifiers were trained only on data for genes belonging to lineages 2 and 3 for MoO. Before training, all genes belonging to 4 genomes from each lineage were removed for them to be used as testing data. From the remaining data, 50% of the genes not labeled as PAV were removed to improve the balance between PAV genes and non-PAV genes in the training data. The model was then trained and tested on the genes from the 8 genomes that were removed before training. The training and testing data split was repeated 100 times to generate average precision, recall, and F1 values as well as average number of true positives, false positives, true negatives, and false negatives for all models.

Feature importances were calculated according to methods described within the rfpimp Python module (https://github.com/parrt/random-forest-importances) (Parr *et al.* 2018). Briefly, a random forest classifier was trained and tested as before to measure a baseline F1 statistic. Each variable in the testing data was then permuted in turn and a new F1 statistic for the model was generated on the permuted data. The difference between the baseline F1 and the new F1 were then calculated. This process was then repeated 100 times and the average decrease in the F1 statistic when each variable was permuted was reported.

Spearman and point biserial correlation coefficients between variables were calculated using the cor function in R version 3.6.1. Phi correlation coefficients were calculated using the psych package (Revelle 2022) version 2.2.9. To calculate dependence statistics for each variable in the complete MoO model, a random forest classifier or a random forest regressor was used to predict each variable originally used to train the PAV gene prediction model using all remaining variables. The same hyperparameters and train–test split were used to train and test each model as for the original PAV gene prediction model. Baseline F1 or $R^2$ values for each model were then calculated and the change in these values when each variable within the model was permuted was calculated as before. However, the results reported were only from a single run of this analysis.

## Data processing and analysis

Data processing was performed in a RedHat Enterprise Linux environment with GNU bash version 4.2.46(20)-release. GNU coreutils version 8.22, GNU grep version 2.20, GNU sed version 4.2.2, gzip version 1.5, and GNU awk version 4.0.2 were all used for processing and handling. Conda (https://docs.conda.io/en/latest/) was used to facilitate installation of software and packages. Code parallelization was performed with GNU parallel (Tange 2018) version 20180322. Previously published data were downloaded using curl version 7.65.3 (https://curl.se/) and sra-tools version 2.10.4 (https://github.com/ncbi/sra-tools). BED format files were processed using BEDtools (Quinlan and Hall 2010) version 2.28.0. VCF format files were processed using bcftools (Danecek *et al.* 2021) version 1.6. SAM and BAM format files were processed using SAMtools (Danecek *et al.* 2021) version 1.8. FASTA format

files were processed using seqtk (https://github.com/lh3/seqtk) version 1.2-r102-dirty.

Data processing and analysis were performed using custom Python scripts (see *Data availability*) written in Python version 3.10.5 with the help of pandas (The pandas development team 2020) version 1.4.3 and numpy (Harris *et al.* 2020) version 1.23.1. GFF format files were parsed in Python using BCBio GFF version 0.6.9 (https://github.com/chapmanb/bcbb/tree/master/gff). FASTA format files were processed in Python using SeqIO from Biopython (Cock *et al.* 2009) version 1.80.

Data processing and analysis were also performed using custom R scripts (see *Data availability*) written in R version 3.6.1 with the help of data.table (Dowle and Srinivasan 2020) version 1.13.6, tidyr (Wickham 2021) version 1.1.3, reshape2 (Wickham 2007) version 1.4.4, and dplyr (Wickham *et al.* 2021) version 1.0.4. Plotting was performed using the ggplot2 package (Wickham 2016) version 3.3.5 and the ggnewscale package (Campitelli 2022) version 0.4.8. Phylogenies were analyzed and plotted using the ape (Paradis and Schliep 2019) package version 5.5 and the phytools package (Revell 2012) version 0.7.90.

## Results

### Genes associated with pathogenicity, non-self-recognition and antibiotic production, are enriched among orthogroups experiencing lineage-differentiating presence–absence variation in *M. oryzae*

Differences in gene PAV events between isolated lineages of *M. oryzae* could be evidence of local adaptation. MoO isolates can be grouped into 4 lineages, called lineages 1, 2, 3, and 4 (Gladieux, Ravel, *et al.* 2018b). Lineages 2, 3, and 4 are monophyletic within the MoO phylogeny and propagate clonally (Gladieux, Ravel, *et al.* 2018b). All lineages show evidence of local adaptation (Thierry *et al.* 2022). To generate a table of all gene PAV events in MoO, we analyzed 123 previously published genomes using a pipeline described in Supplementary Fig. 1a (Zhong *et al.* 2018; Pordel *et al.* 2020; Thierry *et al.* 2022). We first verified that all genomes were of good quality (Supplementary Table 1). Then, these genomes were re-annotated, and the proteomes were clustered into 13,820 orthogroups. We then constructed a phylogeny from a multiple-sequence alignment of all of our SCOs and found each of the 3 clonal MoO lineages formed separate monophyletic groups in our data, as previously observed (Supplementary Fig. 2) (Gladieux, Ravel, *et al.* 2018b; Thierry *et al.* 2022). Using our orthogroups generated from the clustered proteomes, we were able to identify putative orthogroup absences in all genomes. These absences indicated that an entire orthogroup was missing from a genome, rather than single paralogs, which meant that our pipeline was focused on identifying whole-orthogroup PAV rather than within-orthogroup PAV. These putative orthogroup absences were then validated by using TBLASTN (Camacho *et al.* 2009) against the genome and comparing hits to the missing orthogroup using BLASTP (Camachao *et al.* 2009). These validation steps helped ensure that gene absences were not annotation errors. These steps also meant that our approach only counted an absence of an orthogroup when the DNA sequence of the orthogroup was fully missing from a genome and that these absences represented PAV caused by large, gene-sized indels, rather than pseudogenization or gene silencing because of point mutations or small indels.

To identify whether differences in gene PAV events existed between the 3 clonal lineages of MoO, we performed a PCA on our table of PAV events. We found that the top 2 principal components (PCs) of our PCA clearly separated the lineages demonstrating that different PAV events had occurred in each lineage since their separation (Fig. 1a). Next, we identified 587 orthogroups that explained most of the variance in PCs 1 and 2. We called these orthogroups lineage-differentiating PAV orthogroups. We then identified, among all orthogroups, 594 putative effector orthogroups and found, as previously reported (Latorre *et al.* 2020; Thierry *et al.* 2022), that PAV of effector orthogroups alone was sufficient to separate the MoO lineages in a follow-up PCA (Fig. 1b). Given that we identified 4.30% of all orthogroups as putative effectors, the fact that 8.67% of lineage-differentiating PAV orthogroups were effectors represented a clear enrichment ($P < 0.001$, Fisher's exact test). However, non-effector orthogroups still represented 91.33% of lineage-differentiating PAV orthogroups, showing that many orthogroups besides effectors experience lineage-differentiating PAV (Fig. 1c).

To identify what other types of genes were enriched amongst lineage-differentiating PAV orthogroups, we performed gene ontology (GO) and protein family (PFAM) enrichment analysis. This analysis revealed that lineage-differentiating PAV orthogroups were enriched for GO terms related to secondary metabolite production and biosynthesis of membrane components, among other terms (Fig. 2a). Lineage-differentiating PAV orthogroups were enriched for PFAM domains related to antibiotic production, among other domains (Fig. 2b). Genes without PFAM domains were also strongly enriched in lineage-differentiating PAV orthogroups (6,040 annotated, 407 observed, 256.55 expected, $P < 0.001$, Fisher's exact test). Notably, the HET domain, which is associated with heterokaryon incompatibility in fungi, was also enriched among these orthogroups (Fig. 2b). NACHT and NB-ARC domains are characteristic of NOD-like receptors (NLRs) which are involved in fungal self- and non-self-recognition (Dyrka *et al.* 2014). When we grouped these 2 domains together, we found that they were enriched amongst lineage-differentiating PAV orthogroups (23 annotated, 4 observed, 1.15 expected, $P = 0.026$, Fisher's exact test). These results indicated that antibiotic production and non-self-recognition, in addition to effectors, may play an important role in driving adaptation in these 3 isolated lineages of MoO.

### Presence–absence variation genes are more common and spread out throughout the genome in MoT than in MoO

We next sought to identify whether there were specific patterns in the genomic contexts of PAV events in *M. oryzae*. To expand our analyses beyond lineage-differentiating orthogroups and to compare PAV orthogroups with conserved orthogroups, we developed a systematic way to label PAV events. To avoid erroneously calling single gene gain or loss events as actual large-scale variation in presence and absence, we incorporated phylogenetic information in these definitions. Using this information, we identified PAV and conserved orthogroups for each clonal, monophyletic lineage of *M. oryzae*. In our data, orthogroups were labeled as PAV if they were present in all isolates of at least 2 subclades within a lineage and absent in all isolates of at least 2 subclades within a lineage. Subclades were defined as any monophyletic group of isolates that did not include all isolates within the lineage. This definition meant that at least 2 phylogenetically independent loss or gain events needed to be observed in our data for an orthogroup to be labeled PAV. All orthogroups that were present in all but 2 or fewer isolates in a lineage were labeled as conserved orthogroups. All orthogroups that did not fit either definition were labeled as

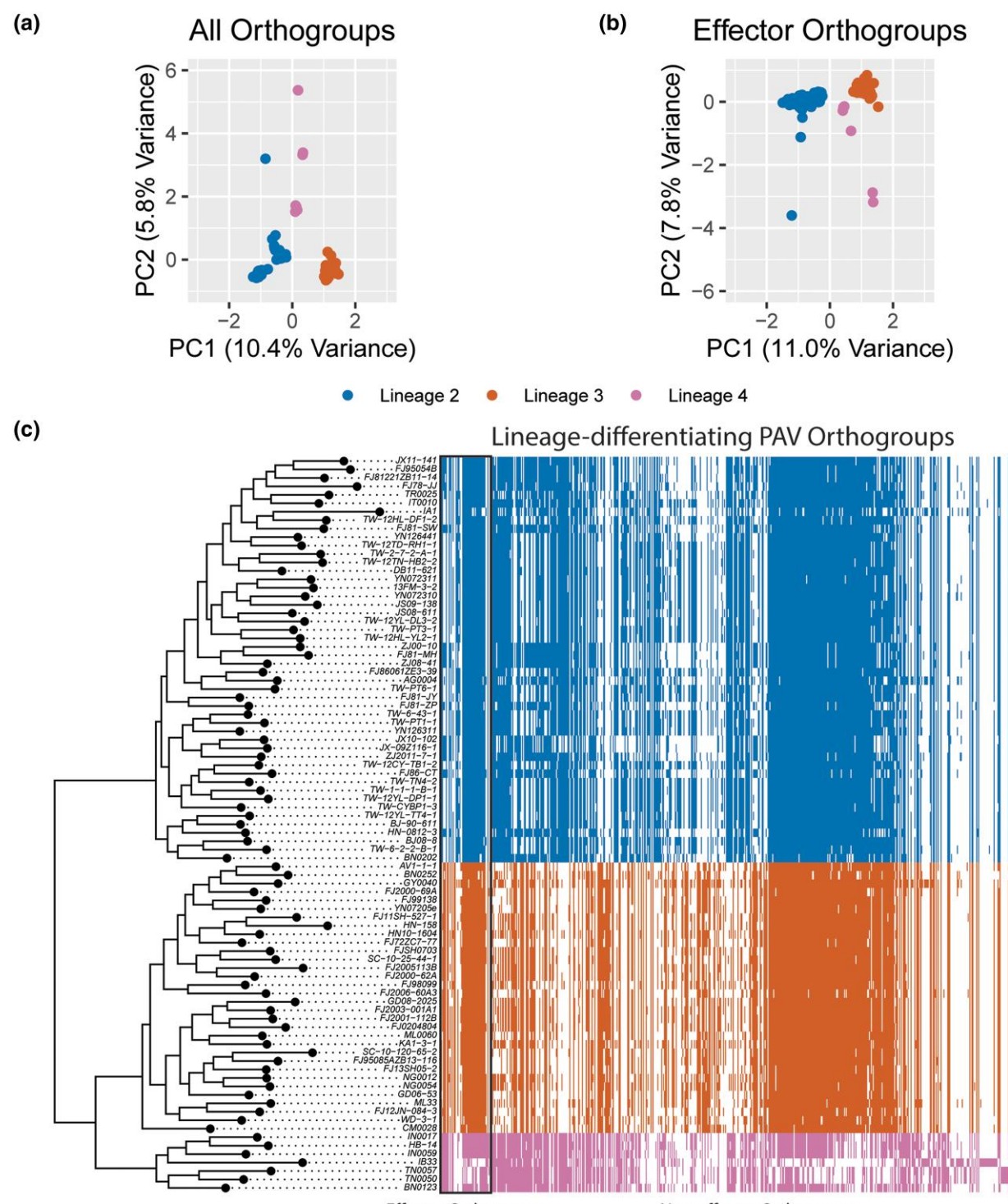

**Fig. 1.** PAV of effector and non-effector orthogroups differentiate the clonal lineages of MoO. a) Scatter plot of values for principal components (PCs) 1 and 2 resulting from a PCA of orthogroup PAV. Each point represents 1 isolate. b) Scatter plot of values for PCs 1 and 2 resulting from a PCA of effector orthogroup PAV. Each point represents 1 isolate. c) Heat map representing which lineage-differentiating PAV orthogroups are present (color) or absent (white) in each genome. Effector orthogroups are separated from non-effector orthogroups by a black box. The phylogeny was generated using a multiple-sequence alignment of SCOs, fasttree and the full MoO phylogeny generated from our data, with lineage 1 omitted (Supplementary Fig. 2). In all panels, colors represent the clonal lineages of MoO. Blue represents lineage 2, orange represents lineage 3, and pink represents lineage 4. Lineages were named as previously described (Gladieux, Ravel, *et al.* 2018b).

"other". Genes belonging to PAV orthogroups or conserved orthogroups were labeled PAV genes and conserved genes, respectively. We describe this pipeline in detail, with examples of what our pipeline labeled as PAV orthogroups in Supplementary

Fig. 1b. This approach allowed us to label 1,269 and 1,029 PAV orthogroups in lineages 2 and 3 of MoO, respectively (Fig. 3a). We did not include lineage 4 in our analysis because of its small number of isolates and omitted lineage 1 because it is thought

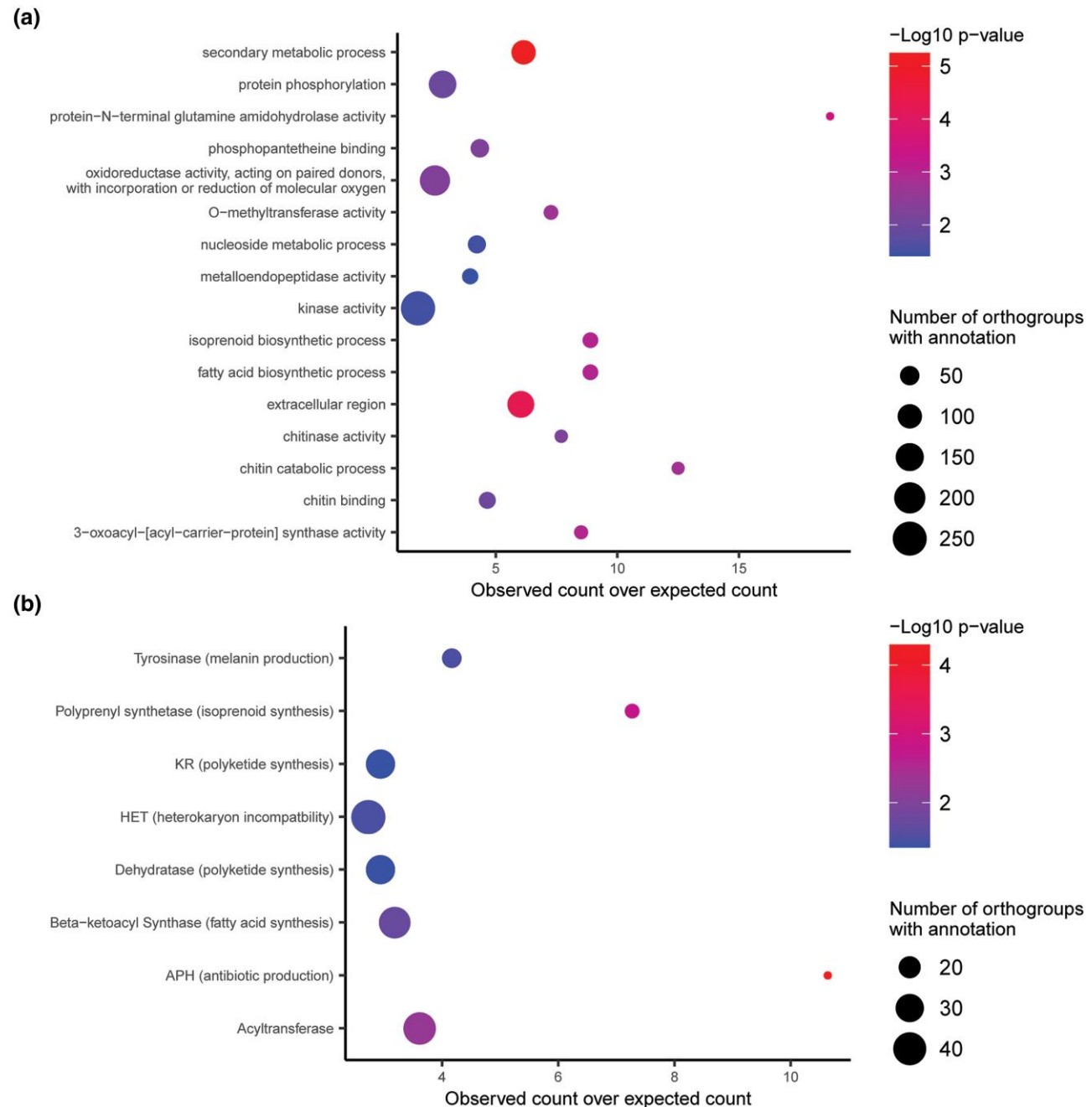

**Fig. 2.** Lineage-differentiating PAV orthogroups in MoO contain many genes related to antibiotic production and non-self-recognition. a) Gene ontology (GO) enrichment analysis of lineage-differentiating PAV orthogroups. b) Protein family (PFAM) domain enrichment analysis of lineage-differentiating PAV orthogroups. P-values shown are the results of Fisher's exact tests. Only GO terms and domains that were assigned to 3 or more lineage-differentiating PAV orthogroups and with enrichment P-values less than 0.05 were reported in this figure.

to be recombining and, in a recombining lineage, it would be impossible to differentiate 2 phylogenetically independent loss or gain events from a single loss or gain event followed by recombination into another monophyletic group (Gladieux, Ravel, et al. 2018b).

To compare PAV across MoO and MoT, we annotated, called orthogroups and validated missing orthologs for 36 previously published MoT genomes. The majority of these genomes were of similar quality to our MoO genomes though some were of higher quality, with a few reaching chromosome and near-chromosome level quality (Supplementary Table 1). Unlike for MoO, only 1

lineage of MoT has been clearly defined. This lineage is a pandemic clonal lineage that recently spread from South America to Asia and Africa (Latorre et al. 2023). Given the small number of MoT genomes available for our analysis compared to MoO, we chose to include other MoT isolates and not to separate them into different lineages for the most of our analyses (Supplementary Fig. 3). All MoT isolates are thought to have propagated clonally since their recent appearance which satisfied the assumptions made by our method for calling PAV orthogroups (Rahnama et al. 2023). In all MoT isolates, we identified 1,570 PAV orthogroups which was substantially more than in the MoO lineages (Fig. 3a).

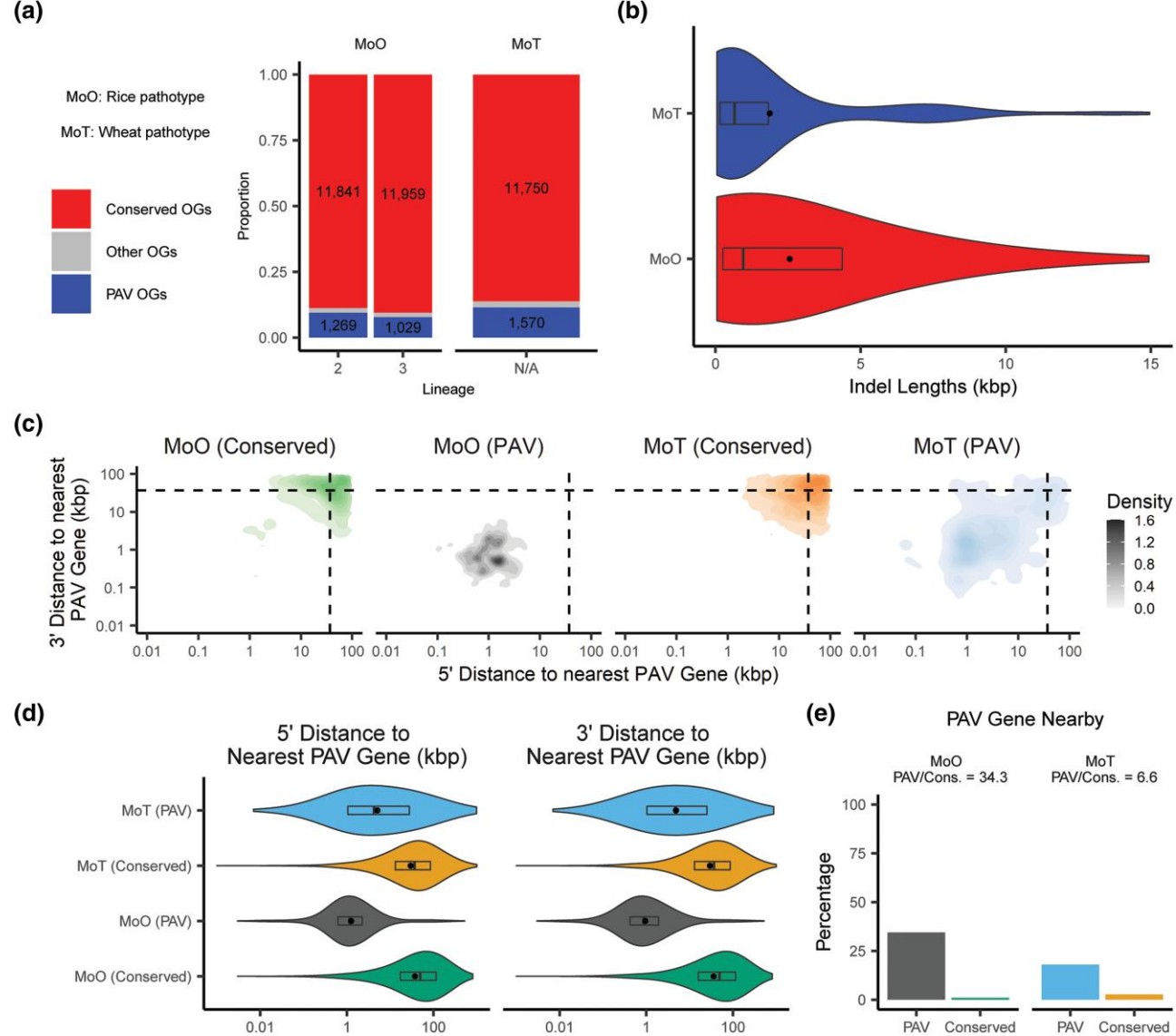

**Fig. 3.** PAV genes are more common and more spread out throughout the genome in MoT than in MoO. a) Stacked barplot comparing the number of PAV orthogroups (OGs) and conserved orthogroups in MoO and MoT. "Other OGs" denote orthogroups that did not satisfy our definitions for either category. b) Distribution of the lengths of large indels (>50 bp) in MoO and MoT. c) Density plot showing the distribution of the distances to the nearest PAV gene for conserved and PAV genes in MoO and MoT. Dashed lines in density plots represent the median values for all genes in both pathotypes. d) Violin plot showing the distribution of the distances to the nearest PAV gene for conserved and PAV genes in MoO and MoT. e) Percentages and proportions of PAV and conserved genes that are within 1,000 bp of a PAV gene in MoO and MoT. Rectangles within violin plots represent interquartile ranges, dark lines represent medians, and dots represent the means with outliers removed. Statistics and statistical comparisons for data shown in panels b) through e) are listed in Supplementary Files 6, 7, 8, 9, and 10.

When we analyzed the isolates belonging to the pandemic clonal lineage alone, we identified only 789 PAV orthogroups. The smaller number of isolates in the clonal pandemic lineage of MoT compared to the MoO lineages (24 isolates vs 32 and 48) as well as differences in evolutionary distances between the MoO and MoT lineages may have influenced these results (Supplementary Figs. 2 and 3).

To assess whether the contrast in the number of PAV orthogroups in MoO and MoT was also present in large indels, we used 117 MoO and 47 MoT Illumina whole-genome sequencing datasets, which represented all lineages of MoO and MoT, to call indels greater than 50 bp in length based on a high-quality reference genome for each pathotype (Supplementary Files 1 and 2). This approach allowed us to identify 1,870 indels in MoO and

1,862 indels in MoT despite using more than double the number of datasets for MoO than MoT (Supplementary Files 4 and 5). We also found that indels were larger in MoO than in MoT, with a median length of 1,818 bp in MoO and 960 bp in MoT (Fig. 3b, Supplementary File 6).

Correspondingly, when we compared the density of PAV genes in MoO and MoT, we found that genes belonging to PAV orthogroups were closer to other genes belonging to PAV orthogroups in MoO than in MoT (Fig. 3c and d, Supplementary File 7). To check whether this result was sensitive to potential differences in genome quality between MoO and MoT isolates, we also used a qualitative measurement of whether or not PAV genes were found within 1,000 bp of conserved and PAV genes in MoO and MoT (Fig. 3e, Supplementary File 8). Again, we found that

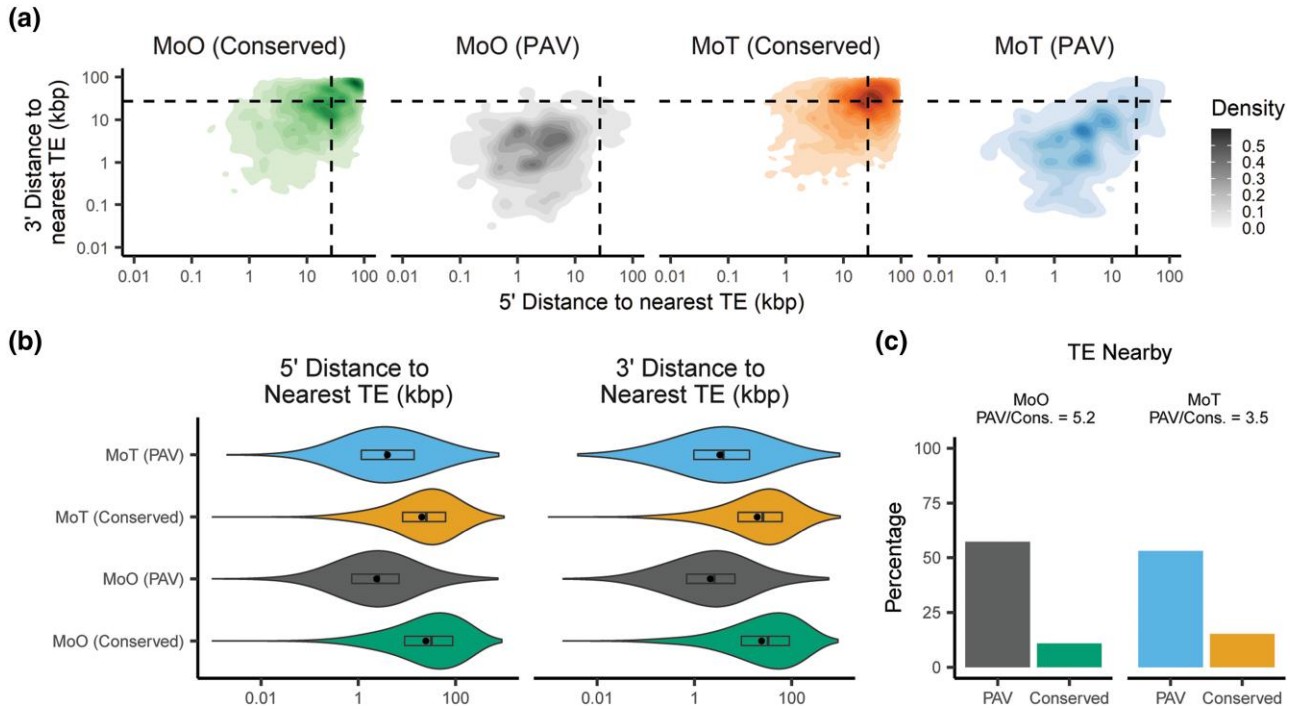

**Fig. 4.** PAV genes are more likely to be found near transposable elements (TEs) than conserved genes. a) Density plots showing the distribution of the distances to the nearest TE for conserved and PAV genes in MoO and MoT. b) Violin plot showing the distribution of the distances to the nearest TE for conserved and PAV genes in MoO and MoT. c) Percentages and proportions of PAV and conserved genes that are within 5,000 bp of a TE in MoO and MoT. Dashed lines in density plots represent the median values for all genes in both pathotypes. Rectangles within violin plots represent interquartile ranges, dark lines represent medians, and dots represent the means with outliers removed. Statistics and statistical comparisons for data shown are listed in Supplementary Files 7, 8, 9, and 10.

PAV genes in MoO were more likely to be found near other PAV genes than in MoT. Taken together, these results indicated that large indels were more likely to involve multiple genes in MoO than in MoT. These results also hinted that gene PAV occurs in regions of the genome that are more isolated from conserved genes in MoO than in MoT.

### Genes prone to presence–absence variation in *M. oryzae* are closer to TEs than other genes

The two-speed genome hypothesis defines 2 genomic compartments in fungal plant pathogens, 1 characterized by rapid evolution, few genes and many TEs, and the other characterized by slow evolution, many genes and few TEs (Dong *et al.* 2015; Torres *et al.* 2020). We investigated whether orthogroups experiencing PAV followed this model in *M. oryzae*. We found that genes in PAV orthogroups were much closer to TEs than genes in conserved orthogroups in both MoO and MoT (Fig. 4a–c, Supplementary Files 9 and 10). While the differences in distance to the nearest gene between conserved and PAV orthogroups in MoO or MoT were typically quite small (median difference < 100 bp), we did find that genes in PAV orthogroups were less likely to be close to genes than conserved genes, though the effect was not as strong as for TEs (Supplementary Fig. 4a–c, Supplementary Files 9 and 10). We also observed differences in these patterns for MoO and MoT. Specifically, we found that PAV orthogroups in MoO were more likely to be close to TEs than those in MoT (Fig. 4c, Supplementary Files 7 and 8). We also found that MoO PAV genes were more likely to be far away from genes than MoT PAV genes (Supplementary Fig. 4c, Supplementary Files 7 and 8).

To understand if these observations also applied to large indels in MoT and MoO, we measured TE and gene densities within the indels we previously identified and within their flanking regions. This analysis revealed that large indels and their flanking regions were enriched in TEs and depleted in genes, though the effect was stronger for TE density than for gene density and validated the fact that our pipeline had identified PAV events generated by large indels in our genomes (Supplementary Fig. 5).

### Genes prone to presence–absence variation exhibit distinct genomic and epigenomic features compared to conserved genes in *M. oryzae*

A previous report has shown that MoO has a much greater TE content than MoT (Nakamoto *et al.* 2023). Given this fact and the increased number of PAV orthogroups in MoT compared to MoO (Fig. 3a), it is unlikely that TEs alone define whether a gene is prone to PAV or not. We therefore chose to investigate whether we could identify other differences in genomic features between PAV genes and conserved genes in *M. oryzae*. We first looked at the GC content of these genes and the regions that flank them. PAV genes were more likely to have lower GC content than conserved genes (Fig. 5a, Supplementary File 11), as did the regions that flank them, though the effect was more subtle for the flanking regions (Supplementary Fig. 6a, Supplementary File 11). We also found that PAV genes were shorter than conserved genes (Fig. 5b, Supplementary File 11). We next performed various functional annotations of PAV and conserved genes and found that PAV genes were more likely to be predicted effectors and less likely to have GO or PFAM annotations than conserved genes (Supplementary Fig. 7, Supplementary File 10).

Next, we gathered histone mark, transcription, methylation, and extrachromosomal circular DNA (eccDNA) sequencing data from the literature to further characterize PAV genes

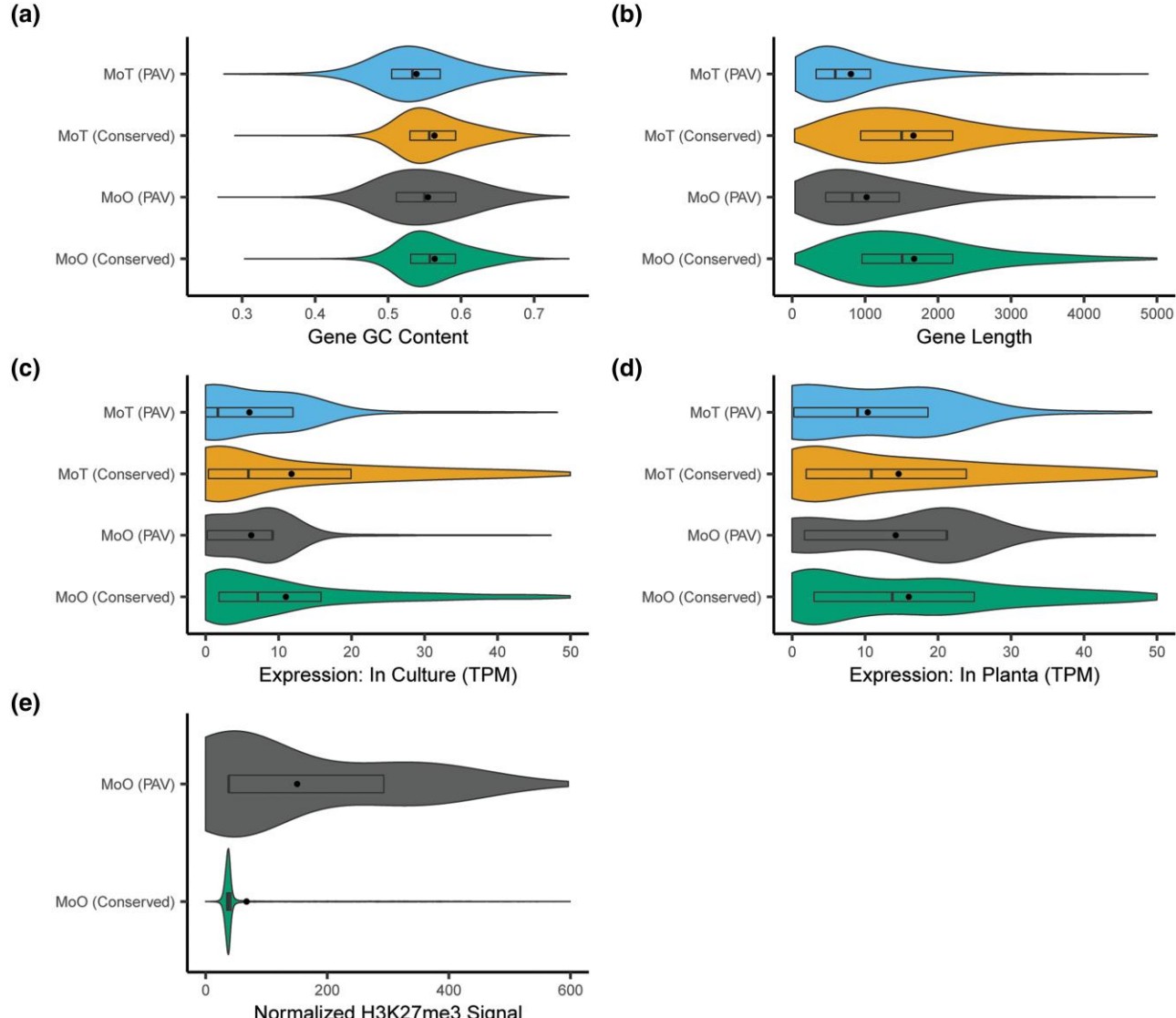

**Fig. 5.** PAV genes are distinct from conserved genes in many ways beyond their proximity to TEs. Violin plots showing the distributions of a) gene GC content, b) gene lengths, c) expression in culture, d) expression in planta, and e) normalized H3K27me3 histone mark ChIP-Seq signal for PAV and conserved genes in MoO and MoT. In panel e), MoT genes were not included as these data are not available for MoT. Rectangles within violin plots represent interquartile ranges, dark lines represent medians, and dots represent the means with outliers removed. Statistics describing the distributions shown and statistical comparisons between these statistics are listed in Supplementary Files 11 and 12.

(Supplementary File 3) (Jeon *et al.* 2015; Peng *et al.* 2019; Zhang *et al.* 2021; Joubert and Krasileva 2022). Unfortunately, these datasets were only available for some strains of MoO or MoT but not for all. Therefore, we first generated values for these features for all genes in 1 reference strain per pathotype. We then averaged these values between orthologs to generate per-orthogroup values and transferred these per-orthogroup values to all other strains. Finally, we used the median value of all orthogroups when an orthogroup was missing from the reference strain. In contrast to MoO, only transcription data were available for MoT, so we were limited to analyzing only expression data for this pathotype. This analysis allowed us to observe that average expression was higher both in culture and in planta for conserved genes than for PAV genes (Fig. 5c and d and Supplementary File 11). Additionally, PAV genes were more likely to show signal from ChIP-Seq of H3K27me3 and H3K36me3 histone marks and less likely to show signal from H3K27ac histone marks (Fig. 5e,

Supplementary Fig. 6b and c, and Supplementary File 11). We also looked at bisulfite sequencing data and found that PAV genes were less methylated and showed a greater variation in methylation percentage than conserved genes (Supplementary Fig. 6d and Supplementary File 11). Finally, we found that PAV genes had a distribution of eccDNA sequencing signal that had a much smaller standard deviation and interquartile range than conserved genes (Supplementary Fig. 6e and Supplementary File 11). Overall, these results indicated clear differences in the genomic and epigenomic features of PAV genes compared to conserved genes.

We found many differences between PAV genes in MoO and MoT including differences in gene length, as PAV genes were smaller in MoT than MoO (Fig. 5b and Supplementary File 12), and differences in expression, as PAV genes in MoT showed lower expression on average than MoO PAV genes both in culture and in planta (Fig. 5c and d and Supplementary File 12). Additionally,

PAV genes were more likely to have GO and PFAM annotations in MoO than in MoT (Supplementary Fig. 7e and f, Supplementary File 8). These observations further supported the idea that PAV may be occurring in different genomic contexts in MoO and MoT.

Finally, we analyzed a similar set of features in the indels we identified in MoT and MoO. We found that these large indels showed decreased GC content, decreased expression, decreased H3K27ac signal, increased H3K27me3 signal and decreased interquartile range in their eccDNA sequencing signal distribution compared to a genomic baseline (Supplementary Fig. 8a–f, Supplementary File 13). We did not observe any difference between large indels and the genomic baseline in median H3K36me3 signal (Supplementary Fig. 8g, Supplementary File 13). We also noticed increased methylation in these regions, which was contrary to our observations in PAV genes (Supplementary Fig. 8h, Supplementary File 13). Overall, these results matched our observations of PAV genes which validated the fact that our pipeline had identified PAV events generated by large indels.

### Genomic and epigenomic features can be used to generate predictive models of gene presence–absence variation in MoO and MoT

Our previous results demonstrated the differences in genomic contexts between PAV genes and conserved genes. We therefore wanted to determine whether these features in aggregate could provide enough information to predict whether a gene was prone to PAV using a machine learning approach. To this end, we trained a random forest classifier on all features we described previously for MoO. We selected this algorithm because of its ease of implementation as well as its robustness to correlated features (Dormann *et al.* 2013). When we trained this model on data from all but 8 strains of MoO and tested the model on the remaining strains, we observed that the model performed very well and was able to predict PAV genes with 86.06% precision and 92.88% recall on average (F1 = 89.34%, Fig. 6a, Supplementary Fig. 9a). Our model also allowed us to determine how important each feature was in predicting PAV genes by calculating the decrease in the F1 statistic when the variable in our testing data was permuted. This approach identified histone H3K27me3 as being the most predictive feature of PAV genes in MoO (Fig. 6b). Feature importances can be influenced by correlations between features (Parr *et al.* 2018). We therefore measured correlations between our features and found that many of them were correlated (Supplementary Fig. 10). We also found that many of these variables could be predicted by other variables (Supplementary Fig. 11). The correlations and dependences between our variables therefore may have influenced the feature importances we observed in our model.

Next, we trained a model to predict PAV genes in MoT using all available data for MoT, which unfortunately did not include histone mark, methylation or eccDNA sequencing data, as previously discussed. Regardless, we found that the model still performed well, with a precision of 94.81% and a recall of 96.43% (F1 = 95.61%, Supplementary Figs. 9b and 12a). In this model, gene expression in planta stood out as being particularly predictive of MoT PAV genes (Fig. 6c). Finally, we trained another MoO model using a reduced set of features that matched the data that was available for MoT, which we called the reduced MoO model. Here we found that the reduced MoO model still performed well with 86.11% precision and 92.21% recall (F1 = 89.05%, Supplementary Figs. 9c and 12b). The similar performance of the 2 MoO models could likely be explained by the high

dependences of our variables, as previously mentioned (Supplementary Figs. 10 and 11). When comparing the reduced MoO model to the MoT model, we noticed some differences between the importances of the features in each model (Fig. 6c and d). For example, in culture expression and the presence of functional annotations was more important in the reduced MoO model than in the MoT model. These differences in importances may have been influenced by the previously described differences in the features of PAV genes in MoO and MoT, including the fact that PAV genes in MoT were less expressed than in MoO and that PAV genes were more likely to have functional annotations in MoO than MoT (Fig. 5c and d, Supplementary Fig. 7, Supplementary Files 8 and 12).

### A predictive model trained on MoT data does not accurately predict presence–absence variation in MoO and vice versa

Finally, we tested if the model trained on MoT data could predict whether genes are prone to PAV in MoO and vice versa. The MoT model performed very poorly on MoO data, with a precision of 25.40% and a recall of 9.06% (F1 = 13.35%, Fig. 6e, Supplementary Fig. 9d). Similarly, the reduced MoO model performed very poorly on the MoT data with a precision of 19.30% and a recall of 9.41% (F1 = 12.65%, Fig. 6f, Supplementary Fig. 9e). This result could be explained by a variety of factors including differences in genomic features between the 2 pathotypes, differences in the importances of each feature in the models, and overfitting of each model. When we analyzed the conserved genes that the MoT model falsely labeled as PAV, we found that many of them were found in isolated regions far away from true PAV genes (Fig. 6g). Similarly, many of the PAV genes in MoT that were not detected by the MoO model were found in isolated regions (Fig. 6h). These results matched our previous observation that genes belonging to PAV orthogroups were closer to other genes belonging to PAV orthogroups in MoO than in MoT (Fig. 3c–e). Our observations, combined with the differences we observed in the genomic and epigenomic features of PAV genes in MoO and MoT described previously, indicated that the patterns and genomic contexts of PAV between the 2 pathotypes are significantly different, despite being within the same species, and indicated that different PAV-generation mechanisms may be acting on these 2 *M. oryzae* pathotypes.

## Discussion

Gene PAV plays an important role in fungal pangenome evolution (McCarthy and Fitzpatrick 2019; Badet *et al.* 2020; Kaushik *et al.* 2022; Moolhuijzen *et al.* 2022). To improve our understanding of these events, we designed a pipeline to identify orthogroups experiencing PAV in *M. oryzae*. We focused our analysis on whole-orthogroup PAV events that were likely the result of large indels, rather than gene silencing or pseudogenization resulting from point mutations. It is therefore important to reiterate that the results of our analyses are only related to these types of events. These analyses should be extended with methods specific to point-mutations and methods that are more suited to track within-orthogroup PAV in the future to see how these results apply to these other kinds of PAV, especially as a greater number of highly contiguous *M. oryzae* genomes become available.

Through our analyses of these PAV events, we found that these events differentiate isolated lineages of MoO, and found that these lineage-differentiating orthogroups are enriched for effectors, as previously published (Latorre *et al.* 2020; Thierry *et al.* 2022). We

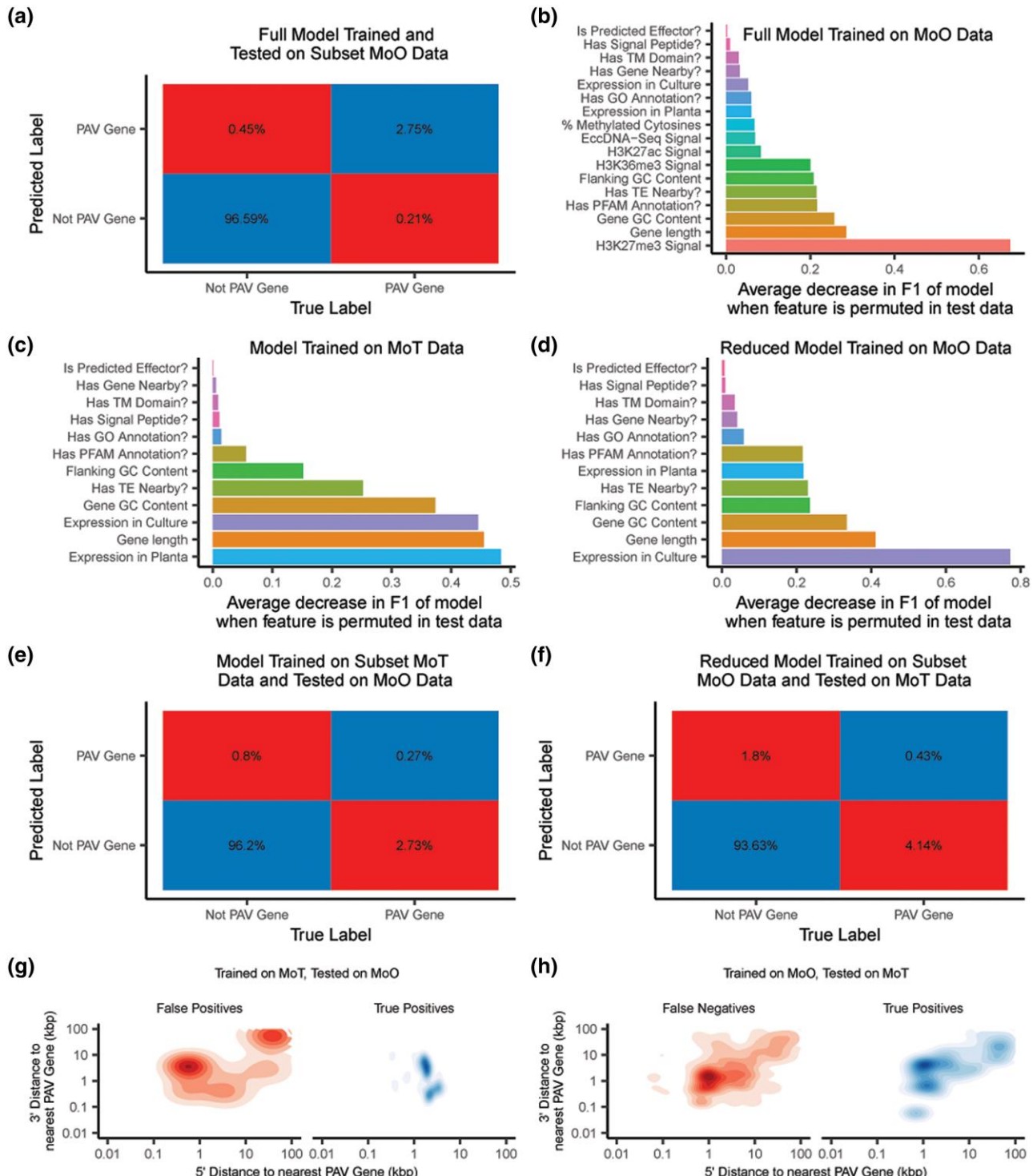

**Fig. 6.** Random forest classifiers accurately identify PAV genes in MoO and MoT, but the models perform poorly on genes from the host they were not trained on. a) Confusion matrix showing average percentages for each classification outcome of the MoO random forest classifier when tested on MoO genes that it was not trained on. b) Decrease in the F1 statistic of the MoO random forest classifier when each feature is permuted in the testing data. Features described as questions are binary, all other features are continuous. c) Decrease in the F1 statistic of the MoT random forest classifier when each feature is permuted in the testing data. d) Decrease in the F1 statistic of the MoO random forest classifier trained on a subset of features (reduced MoO model) when each variable is permuted in the testing data. e) Confusion matrix showing average percentages for each classification outcome of the MoT random forest classifier when tested on MoO genes. f) Confusion matrix showing average percentages for each classification outcome of the MoO random forest classifier trained on a subset of features (reduced MoO model) when tested on MoT genes. g) Density plots showing the distribution of the distances to the nearest PAV gene for false positive and true positive predictions by the MoT random forest classifier when tested on MoO genes. h) Density plots showing the distribution of the distances to the nearest PAV gene for false negative and true positive predictions by the MoO random forest classifier trained on a subset of features (reduced MoO model) when tested on MoT genes.

also found that genes related to both antibiotic production and non-self-recognition were enriched among lineage-differentiating PAV orthogroups. Given that multiple enriched PFAM and GO terms were related to these functions, and that these functions included both defense against infection and external inhibition of bacterial growth, these results could point to the local and rice-associated microbiome playing an important role in *M. oryzae*'s evolution. All 3 clonal lineages are geographically isolated and experience different climates (Thierry *et al.* 2022). They also tend to infect different rice varieties and cause disease of varying severity (Thierry *et al.* 2022). Geography and host genotype could have major influences on the microbiome the fungus encounters as different rice genotypes have been found to have different microbiomes (Xiong *et al.* 2021; Singh *et al.* 2022; Zhang *et al.* 2022). Further microbiome sampling of rice varieties used in these areas as well as the environmental microbiome could therefore give better insight into the results we present here and how these microbiomes might shape the fungus' fitness. Additionally, it is important to note that adaptation to the host microbiome and the environment in general are often forgotten when discussing fungal plant pathogen evolution. Our results point to the importance of considering these factors when studying the success of these pathogens. Unfortunately, we could not extend these analyses to MoT as lineages of MoT have not been as thoroughly characterized as MoO and therefore detailed information on their geography or host phenotypes is not yet available.

We then looked to find features of PAV orthogroups that might help us better understand where these events are occurring in the genome. We found that these events were associated with a high TE density and a low gene density, though the effect was stronger for TE density than gene density. We also found that PAV genes are shorter, have lower GC content, and are more likely to be effectors. Finally, PAV genes are less expressed and display stronger histone H3K27me3 signal than conserved genes. When we combined all of these features into a predictive model, we found that the model performed very well and predicted PAV genes with 86.06% precision and 92.88% recall, on average. We were also able to identify histone H3K27me3 as the most predictive feature, though gene length and GC content stood out as well. Although prediction accuracy by random forest classifiers is robust to correlated features (Dormann *et al.* 2013), the variable importances we observed were likely influenced by the fact that several variables in our model were correlated with each other and that many showed high dependences, which meant that the information encoded in these variables could also be described by other variables in the model. These importances should therefore be interpreted with caution. Through reanalyzing previously published Illumina sequencing data, we were able to directly identify large indels in our *M. oryzae* isolates and compare their genomic features to those of the rest of the genome. We found that this structural variation occurred frequently in TE-dense and gene-sparse areas of the genome, and that GC content, RNAseq signal, and H3K27me3 ChIP-Seq signal for these regions resembled that of our PAV genes. This result was evidence that our pipeline was successful in specifically identifying PAV events that were generated by large indels.

Many of the features that were particularly important in our classifier were related to the two-speed genome concept which supported the idea that gene PAV in *M. oryzae* is strongly associated with the rapidly evolving compartment of the genome (Dong *et al.* 2015; Torres *et al.* 2020). Our findings support the idea that these features may play an important role in the evolution of the pathogen and reflect previous findings on the

association between TEs and the evolution of the accessory portion of fungal pangenomes (Badet *et al.* 2020). However, the fact that the presence of TEs were important features in our random forest classifier but not amongst the most important, supports the idea that the relationship between TEs and rapid evolution is not always a causal one and that complex correlations are at play. In short, other variables may be shaping the PAV-prone compartment of the *M. oryzae* genome and driving both rapid evolution and TE activity. This is also supported by the fact that, while TEs are more common in MoO (Nakamoto *et al.* 2023), we found that MoT appears to be experiencing PAV at a faster rate than MoO.

While the gene space for the genomes we analyzed were well assembled, most of the genomes we performed our analysis on were not chromosome-level assemblies. Therefore, although we observed features associated with subterminal regions in our PAV genes, such as the association of PAV genes with TEs, we could not confirm previous findings on the association between subterminal regions and accessory genes in other fungi (McCarthy and Fitzpatrick 2019). This analysis should be repeated once more high-quality genomes become available for *M. oryzae* to fully determine whether these findings apply to the blast fungus as well. Similarly, though the genomic and epigenomic features associated with PAV that we identified in this study should be kept in mind when studying other pangenomes, it is unclear whether the features of gene PAV we identified are applicable to other fungi, and therefore more in-depth studies of these genomic and epigenomic features are necessary to assess how broad these findings are as datasets become available for more fungi. Likewise, though our random forest classifier performed well, many of the data we used were only available for a reference *M. oryzae* isolate which likely affected how we ranked the importances of each variable in our model. To validate our results, our approach would need to be repeated using data for each isolate. However, a model using a subset of our features performed well, indicating that RNAseq data for each isolate may be enough to substantiate these results.

Our model showed that PAV genes can be identified using features in the genome, therefore establishing a method to identify genes prone to PAV in *M. oryzae* without relying on phylogenetics. This method could therefore be useful for identifying genes prone to PAV in lineages of MoO with very few isolates, like lineage 4, or for studying PAV in groups of genes with complicated evolutionary relationships like sequence unrelated structurally similar (SUSS) effectors (Seong and Krasileva 2021). While our models performed well, they also identified genes that had features of PAV genes but did not experience PAV. For example, our model trained on the full MoO dataset marked 0.45% of genes as PAV in our testing data when they were conserved. These false positives could be caused by incomplete sampling of *M. oryzae* isolates in our dataset and therefore larger datasets should be considered to verify that these are biologically meaningful observations. If that were the case, these false positives could help us better understand which genes are under strong selection to be kept in the *M. oryzae* genome or which genes' genomic contexts are changing to look more like conserved genes. Notably, our results support the exciting possibility of using genomics to predict targets for disease-prevention strategies that will remain in the genome, therefore making these strategies more robust.

Finally, we found distinct patterns in the genomic contexts of PAV genes in MoO and MoT. Specifically, we found that PAV in MoT was more common and spread out throughout the genome than in MoO. The differences in evolutionary distances between isolates in the lineages of the 2 pathotypes as well as differences

in the number of isolates in each lineage may have contributed to the differences in the number of PAV orthogroups. Specifically, these differences may have contributed to the significant decrease in PAV orthogroups when we analyzed the clonal pandemic lineage of MoT alone. However, supporting evidence from our analysis of large indels, in which we analyzed sequencing data from all lineages of MoO and MoT, support our findings of increased PAV in MoT compared to MoO. We also found that many of the genomic and epigenomic features of PAV that we identified in MoO were different in MoT. These differences may have explained why our MoO random forest classifier performed poorly on MoT data and vice versa, since the patterns in the false positives and false negatives of these tests reflected the observed differences in PAV between MoO and MoT. These results in aggregate hint at differences in the evolution of the rice and wheat pathotypes of *M. oryzae* and especially differences in the mechanisms that generate PAV events in the 2 pathotypes.

The 2 *M. oryzae* pathotypes share some major differences in their TE content (Nakamoto *et al.* 2023) and very different life histories, with MoO originating 9,800 years ago (Gladieux, Ravel, *et al.* 2018b) and propagating mostly clonally since then, while MoT is thought to have emerged approximately 60 years ago from a multi-hybrid swarm of many different *M. oryzae* pathotypes (Ceresini *et al.* 2019; Rahnama *et al.* 2023). We propose that the differences in PAV across the 2 pathotypes reflect these life histories, with MoO exhibiting more of a stable equilibrium and much slower paced evolution, where PAV events happen in specifically defined compartments of the genome, while MoT is rapidly losing and gaining genes, even in areas of the genome where most of the conserved genes in MoO are located. It is unclear at this point whether MoT is heading toward an equilibrium that will resemble MoO, or whether there are key differences between the 2 pathotypes that are shaping their genomes beyond their evolutionary histories. MoT, which appears to be evolving more rapidly than MoO, will pose a significant challenge for disease prevention. A better understanding of these evolutionary dynamics and the differences between MoO and MoT could help us better comprehend why MoT is such a devastating emerging pathogen and help us curb its threat. Finally, these results highlight the need to study isolated populations of a species separately as well as in aggregate to understand whether observations made for the pangenome applies to every population within a species, especially if they are adapted to different hosts or environments and if they have distinct evolutionary histories. It also highlights the fact that multiple types of PAV may be occurring in different populations of the same species and that more sensitive comparison of PAV events, for example those occurring due to various sizes of indels, may be necessary in the future. These patterns in the genomic features associated with PAV could also be difficult to compare across species as different PAV mechanisms could be associated with distinct features.

Our study demonstrates that gene PAV can be associated with specific genomic and epigenomic features in fungi and that these associations can be predictive. We also show that major variation can exist in these features between different populations of the same species and that distinct mechanisms might generate PAV in these populations. This study highlights the need for more research on fungal pangenomes and the genomic features and PAV mechanisms that define them to better understand how fungi adapt to their environments. These studies could also lead to a greater understanding of how fungal plant pathogens adapt to their hosts and predicting these adaptations could help us develop more effective disease-prevention strategies. We propose that considering intra-species variation and evolutionary history of different populations is important to fully capture potential variations in PAV-generation mechanisms within the species.

## Data availability

All code for data generation, analysis, and plotting is available on GitHub: https://github.com/pierrj/moryzae_pav_manuscript_code. All files used for analysis and plotting are available on Zenodo under the DOI 10.5281/zenodo.7444379.

Supplemental material available at GENETICS online.

## Acknowledgments

We thank Krasileva lab members for thoughtful comments and feedback on the manuscript. This research used the Savio computational cluster resource provided by the Berkeley Research Computing program at the University of California, Berkeley (supported by the UC Berkeley Chancellor, Vice Chancellor for Research, and Chief Information Officer).

## Funding

PMJ has been supported by the Grace Kase-Tsujimoto Graduate Fellowship and the National Institute of Health New Innovator Director's Award (https://commonfund.nih.gov/newinnovator), grant number DP2AT011967. KVK has been supported by funding from the Innovative Genomics Institute (https://innovativegenomics.org/), the Gordon and Betty Moore Foundation (https://www.moore.org/), grant number 8802, and the National Institute of Health New Innovator Director's Award, grant number DP2AT011967. The funders had no role in study design, data collection and analysis, decision to publish, or preparation of the manuscript.

## Conflicts of interest

The author(s) declare no conflicts of interest.

## Author contributions

PMJ conceptualized and designed the study with input from KVK. PMJ designed the analyses and analyzed the data. PMJ wrote the original draft manuscript. PMJ and KVK reviewed and edited the manuscript. Both authors read and approved the final manuscript.

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
