## [Peer Review File · Genetics]

Distinct genomic contexts predict gene presence-absence variation in different pathotypes of *Magnaporthe oryzae*

Pierre Joubert and Ksenia Krasileva

NOTE: The reviews and decision letters are unedited and appear as submitted by the reviewers.

In extremely rare instances and as determined by a Senior Editor or the EIC, portions of a review may be redacted. If a review is signed, the reviewer has agreed to no longer remain anonymous.

The review history appears in chronological order.

Review Timeline:

Submission Date:	2023-03-10
Editorial Decision:	2023-05-12
Resubmission Received:	2023-08-14
Editorial Decision:	2023-09-19
Resubmission Received:	2023-11-28
Accepted:	2023-12-19

May 12, 2023

GENETICS-2023-305992

Distinct genomic contexts predict gene presence-absence variation in different pathotypes of a fungal plant pathogen

Dear Dr. Krasileva:

Two experts in the field have reviewed your manuscript, and I have read it as well. The approaches and findings to incorporate the gene content variation in the population genomics of this important plant pathogen compelling and worthy of publication. While your manuscript is not currently acceptable for publication in GENETICS, we would welcome a revised manuscript. Both reviewers have comments and concerns to be addressed in a revised manuscript. You can read their reviews at the end of this email.

These are aspects raised by the reviewers that suggest a more broad contextual view of the Mo pangenome is warranted. Both also indicate a treatment of potential mechanisms of PAV be discussed and examined in these genome data. While of course direct detection of PAV mechanism is not going to be possible a little more treatment of the potential drivers of this is requested. The concept of inparalog patterns among orthologous groups is also raised, I was not sure if the authors took this into account or restricted analyses on unambiguous single copy orthologs or if synteny was considered in some of the assignments. If maybe possible to incorporate additional filters to the data if needed (eg the tool genespace does use synteny among assembled genomes to further refine orthofinder orthology clusters). I'm not sure if this is needed but providing some ideas around the details raised by the reviewers. Overall you should be able to address these concerns with some additional refinement in analysis and presentation but I leave decisions as to how to best address the concerns to the authors who know the nuance of the dataset best.

We look forward to receiving your revised manuscript. Please let the editorial office know approximately how long you expect to need for revisions.

Upon resubmission, please include:

1. A clean version of your manuscript;
2. A marked version of your manuscript in which you highlight significant revisions carried out in response to the major points raised by the editor/reviewers (track changes is acceptable if preferred);
3. A detailed response to the editor's/reviewers' feedback and to the concerns listed above. Please reference line numbers in this response to aid the editor and reviewers.

Your paper will likely be sent back out for review.

Additionally, please ensure that your resubmission is formatted for GENETICS
<https://academic.oup.com/genetics/pages/general-instructions>

Follow this link to submit the revised manuscript: <https://genetics.msubmit.net/cgi-bin/main.plex?el=A5NR5FSR1A4UsF3I3A9ftdx8UqbWYXhrTT0dmi1C8EowZ>

Sincerely,

Jason Stajich
Associate Editor
GENETICS

Approved by:
David Begun
Senior Editor
GENETICS

Reviewer #1 (Comments for the Authors (Required)):

Review Joubert et al, 2023

Overview of the study

With the recent sequencing efforts aiming at providing whole-genome assemblies for multiple individuals of a same species, it has become evident that single-reference genomes are often not capturing the extent of genetic diversity present within a species. Fungal plant pathogens in particular have been shown to harbour large amounts of accessory genes, i.e genes displaying presence-absence variation (PAV) within the species. Identifying the determinants of gene PAV is therefore crucial to better understand what drives species' pangenome evolution.

In their study, Joubert & Krasileva explore the (epi)genomic features that might underly such gene presence-absence variation (PAV) in two pathotypes of the fungal plant pathogen

responsible for the blast disease on rice and wheat (abbreviated MoO and MoT). The major conclusion they reach is that in the two considered pathotypes, machine-learning can accurately predict which genes show PAV in each pathotype. In addition, they highlight that gene PAV associates with different genomic environments in both pathotypes.

For that purpose they first establish a map of gene PAV in genomes of rice and wheat-infecting isolates. They first show that patterns of gene PAV in rice-infecting isolates delineates known clonal lineages of the pathogen (Figure 1). Using predicted gene ontologies (GO) and PFAM annotations, they highlight that specific protein functions are more prone to PAV (Figure 2).

Next, they focus on two clonal lineages of the rice-infecting pathotype (MoO) and two of the wheat-infecting pathotype (MoT). Applying a phylogeny-informed method to call gene PAV, they conclude that the wheat-infecting pathotype (MoT) is more prone to PAV than the rice-infecting pathotype (MoO) (Figure 3). Looking at the genomic features that associate with gene PAV, they highlight contrasted genomic environments for genes showing PAV (Figure 3 and 4). Gene length and GC content in particular were found to differ between conserved and genes showing PAV (Figure 5).

To formally link genomic features to gene PAV, the authors implement a machine-learning algorithm to classify genes as conserved or displaying PAV. Major predictors of gene PAV include proximity to transposable elements, histone methylation marks and gene length together with gene expression. When trained and tested on the same pathotype, the models were able to accurately assign most of the gene set to the correct category (Figure 6). Finally, the authors end by highlighting the inaccuracy of the individual models to classify gene PAV in the alternative pathotype.

Major comments

The current study adopts an original angle to address a timely question : what drives pangenome evolution. Overall, the analysis are sound and the methods properly described, providing interesting insights into the evolutionary dynamics at place in a major plant pathogen.

My major comment is regarding to what the authors define as gene PAV along the manuscript. I would clarify what type of mechanisms might be underlying gene PAV in the current dataset. Is for instance pseudogenization considered gene PAV in the current analysis? One would expect contrasted genomic signatures for different mechanisms leading to gene PAV (point mutations or large indels), which should be discussed in light of the machine-learning results.

In addition, gene PAV are defined here at the orthogroup level, but single orthogroups can count multiple paralogs that also show PAV. I believe providing a more general view of what the pangenome looks like in terms of the proportions of core (gene present in all isolates), accessory (all but one isolate), multi-copy (paralogs) and unique (single isolate) orthogroups in the two pathotypes could help clarifying the results. In particular looking at the shape of the gene PAV accumulation curve for varying number of sampled genomes (addressing sampling bias).

Minor comments

Through the manuscript I found the reading a bit complicated sometimes. Shorter and more direct sentences could help. In addition, the back and forth between MoO / MoT, the clonal / non-clonal lineages within each pathotype and different methods to detect PAV complicate the story. Reorganising some of the paragraphs could help improving the manuscript readability.

Introduction

I believe the introduction could benefit from a bit more context on the *Magnaporthe oryzae* species (syn. *Pyricularia oryzae*, without entering the taxonomic debate) as it has a complex history linked to plant host range / specificity and clonality / hybridization. It would set the stage when transposing the major conclusions in the context of other species (later in the discussion).

L9-10 > confusing

L14 > syn. *Pyricularia oryzae*, without entering the taxonomic debate but for clarity

L22 > does it mean that the prediction didn't work for the other species?

L29 > not all microbes have expansive pangenomes (i.e closed pangenomes)

Specify that pangenome here is considered as the set of genes, and not as the alternative definition of the set of sequences (coding and non-coding).

L32 > not sure the reference supports the claim, evidence for HGT plus recent large transposons that associate with large genomic regions (starships).

L35 > pan-genome or pangenome, consistency across manuscript

L54-55 > facilitate how?

L56 > No mention of the recent history of the pathogen (without focussing too much on the taxonomic debate but worth mentioning? Species complex or lineages with varying host specificity / range, potential for hybridization?

L68 > Related to above, clarify what is referred as "species" here, so that it is clear how it might or not relate to any other species.

L74 > be more specific?

Results

I missed a bit of context for the analysis of structural variants based on short-reads. I would clarify how gene PAV and indels can be linked to each other (gene PAV likely generated through indels). Mentioning the proportion of indels that overlap/associate with genes could be informative for instance. Note that as presented I'm not sure one can differentiate deletions from insertions, hence indels.

L88-89 > Supplementary table with pooled metadata for the genomes could be useful (host, lineage, genome metrics, assembly ID ..)?

L92 > nor putative pseudogenes?

L98 > reference to the methods section?

L102 > lineage 4 seems closer to lineage 3 with effectors than with all PAV on PC1. Curious about how does a Venn diagram or upset plot looks like for all PAV and effectors only.

L102 > fisher test?

L114 > subset for all isolates from lineage 2, 3 and 4 I presume?

L119 > reference to the methods section? Why ignore lineage specific PAV here?

L124-126 > Lineage-differentiating PAV orthogroups?

L134 > rice-infecting here, MoO before, consistency

P138 > a bit confusing here what was intended and the corresponding reassignment of gene PAV across MoO lineages. Schematic could help? Or presenting the number of genes that are : monophyletic present/absent (excluded in current analysis), present/absent in at least two distinct groups, or missing in 1 genome only, could be useful to clarify the pipeline.

L154-155 > not clear why the recombining lineage would violate assumptions and was excluded

L165 > "in in".

L168 > Supplementary table with pooled genomes metadata (host, lineage, genome metrics, assembly ID ..)?

L172-173 > confusing, "datasets of comparable size for further comparisons"

L177-178 > Add reference to the method section?

L176 > indels rather than deletions?

L179 > would the average number of indels per genome/isolate make the statement more compelling?

L183 > Label for Figure 3C missing. Figure 3C and others heatmaps lack a scale. "Much more likely", report the statistical test in the text and figures too?

L185 > No mention so far of what is the proportion of indels that overlap/associate with genes. Would clarify the link

L186 > Not clear what defined regions are here. Larger indels promote the emergence of gene PAV clusters in MoO?

L193 > Figure 4A-C

L192-200 > provide results from statistical tests in text and figures too?

L206 > Figure S4, not clear what the y axis represents given the scale and curve shape (does not seem like gene/TE counts, rather a scaled density over genomic windows? Not clear why it caps at 1 for genes but goes over 1.5 for TEs?). The methods section mention 10 bp windows, implying that for most bins coverage values are binary (either 0 or 1). Showing smoothed estimates (+ standard error) over longer genomic windows could provide a more direct representation of the genomic context for indels. Briefly describing (in the methods?) the scaling method (deeptools) used here could help clarify the figure. Adding legend to the plot itself (next to coloured lines) would facilitate the reading I think.

L215-216 > "and some differences in these features" reads vague

L236-237 > Figure 5 (and supplementary), the density plots make differences hard to visualize. Violin plots as an alternative? Add statistical annotation to the figure itself?

L242 > Unclear what was done here

L250 > unclear what "tighter signal distribution" means here

L262-263 > above it is stated that "average expression was higher both in culture and in planta for conserved genes than for PAV genes", correct?

L266 > Phrasing. "Genomic and epigenomic features can be used to generate predictive models of gene presence-absence variation in rice and wheat-infecting *M. oryzae*"?

L278-281 > long sentence that reads more like discussion. Confusing what the values in Figure S9 represent (decrease in F1 or R2 metrics, so that the higher the value the stronger the predictor?)

L284 > clarify what is meant by reduced model (not always consistent in text and figures). Some variables such as epigenetic marks not available for MoT?

L293 > specify which differences

L318-320 > Providing raw numbers of true/false positives/negatives in confusion matrices would help clarify the point (Figure 6 and Figure S10)

L323 > recapitulate which are the patterns

Discussion

As for the introduction, I missed some information related to the evolutionary history of *M. oryzae* as a fungal pathogen. In addition, the authors mention strong differences in TE content between pathotypes, any explanations in light of their results (i.e MoO bears more TEs but experience less PAV compared to MoT)?

L389 > these could well be true positives not detected given the current sampling.

Methods

Methods are well presented and all the data generated in the manuscript together with the associated scripts are fully available online, I really appreciate it.

L457-463 > what was the set size for this preliminary fine-tuning step?

L464 > intermediary step to convert tblastn output to gff, additional AGAT options?

L467 > not clear to me

L564 > Not clear what "profile plot" refers to. "Window-based coverage calculation of genes and transposons"?

L579-580 > Got confused by the phrasing, all genomes from lineages 2 and 3 except 2 times 4 genomes were used as the training set and the 8 excluded genomes were then used for testing, correct?

Reviewer #2 (Comments for the Authors (Required)):

In this study, the authors look at gene presence-absence polymorphism in different pathotypes of an important fungal plant pathogen using publicly available genomic data.

I have three major comments:

1) the authors didn't mention whether they consider or not the dispensable chromosomes previously described in the studied species; substantial gene presence-absence polymorphism is expected on dispensable chromosomes and comparison between core and dispensable chromosomes could be interesting ; if the author focus on core chromosomes only it should be explained to interpret this study

2) I didn't understand well the gene enrichment analyses results; the HET gene and genes related to antibiotic production didn't appear much more enriched than other gene categories ; in any case, I found the discussion about microbiome importance very speculative as finding genes related to antibiotic production doesn't imply necessarily a link with antibiotic production; if it is the case, references should be provided

3) information on quality of genome assemblies (N50, L50, etc..) and genome coverage of Illumina reads used for deletion detection should be given to assess the power of detecting PAV in all strains; in particular differences described between the pathotypes might be due to technical aspects only

I have the following minor comments:

1) the introduction could be reframed as the authors highlight some limits of previous studies which in my opinion were actually addressed in these studies; see reference 6 as an example; the authors can present their study with other arguments

2) statistical testing is lacking for several analyses (ex: L103-106)

- 3) L147: it is unclear what is a subclade in a lineage and how the author defines it
- 4) L310 : subtitle is too long
- 5) L370 : it is unclear what are subterminal regions and what link did the author find
- 6) L388 : "many genes" is not informative

Two experts in the field have reviewed your manuscript, and I have read it as well. The approaches and findings to incorporate the gene content variation in the population genomics of this important plant pathogen compelling and worthy of publication. While your manuscript is not currently acceptable for publication in GENETICS, we would welcome a revised manuscript. Both reviewers have comments and concerns to be addressed in a revised manuscript. You can read their reviews at the end of this email.

These are aspects raised by the reviewers that suggest a more broad contextual view of the Mo pangenome is warranted. Both also indicate a treatment of potential mechanisms of PAV be discussed and examined in these genome data. While of course direct detection of PAV mechanism is not going to be possible a little more treatment of the potential drivers of this is requested. The concept of inparalog patterns among orthologous groups is also raised, I was not sure if the authors took this into account or restricted analyses on unambiguous single copy orthologs or if synteny was considered in some of the assignments. If maybe possible to incorporate additional filters to the data if needed (eg the tool genespace does use synteny among assembled genomes to further refine orthofinder orthology clusters). I'm not sure if this is needed but providing some ideas around the details raised by the reviewers. Overall you should be able to address these concerns with some additional refinement in analysis and presentation but I leave decisions as to how to best address the concerns to the authors who know the nuance of the dataset best.

We have incorporated the reviewers feedback in a new draft of the manuscript. We have rewritten many parts of the Introduction, Results and Discussion, to clarify that our manuscript focuses on whole-orthogroup PAV caused by insertions and deletions of DNA of different length (indels), rather than within-orthogroup copy number variation or those events caused by point mutations. We have addressed the implications of this choice in the manuscript as well as in this detailed response to the reviewers.

We have also made our statistics more clear and obvious to the readers throughout the manuscript, and clarified our use of the terms we define within the manuscript to be more consistent and easier to understand.

As previously mentioned in a communication to the editor during the review process, we have also fixed a mistake in our manuscript which we discovered after submission. We uncovered improper labeling of many MoT genomes which originated from a mistake in our code used to download these genomes. 52 out of 88 MoT genomes we used in our initial analysis were unfortunately not wheat-infecting, but instead infected other hosts. We have removed these genomes from our analysis. This meant that instead of two MoT lineages, we only analyzed a single lineage. Fortunately, this change has not affected the majority of our results, and has strengthened many of them (especially differences in genomic features and PAV density of MoT vs MoO). The only major result that was affected by this change was the number of PAV orthogroups in MoT vs MoO. Previously, we observed two to four times more PAV orthogroups in MoT, we now only observe 50% more. We believe that these changes do not detract from the

results we present in the manuscript. We updated our code on github and STables to ensure dataset information is fully available for reproducibility.

We look forward to receiving your revised manuscript. Please let the editorial office know approximately how long you expect to need for revisions.

Upon resubmission, please include:

1. A clean version of your manuscript;
2. A marked version of your manuscript in which you highlight significant revisions carried out in response to the major points raised by the editor/reviewers (track changes is acceptable if preferred);
3. A detailed response to the editor's/reviewers' feedback and to the concerns listed above. Please reference line numbers in this response to aid the editor and reviewers.

Your paper will likely be sent back out for review.

Additionally, please ensure that your resubmission is formatted for GENETICS
<https://academic.oup.com/genetics/pages/general-instructions>

Follow this link to submit the revised manuscript: <https://genetics.msubmit.net/cgi-bin/main.plex?el=A5NR5FSR1A4UsF3I3A9ftdx8UqbWYXhrTT0dmi1C8EowZ>

Sincerely,

Jason Stajich

Associate Editor

GENETICS

Approved by:

David Begun

Senior Editor

GENETICS

Reviewer #1 (Comments for the Authors (Required)):

Review Joubert et al, 2023

Overview of the study

With the recent sequencing efforts aiming at providing whole-genome assemblies for multiple individuals of a same species, it has become evident that single-reference genomes are often not capturing the extent of genetic diversity present within a species. Fungal plant pathogens in particular have been shown to harbour large amounts of accessory genes, i.e genes displaying presence-absence variation (PAV) within the species. Identifying the determinants of gene PAV is therefore crucial to better understand what drives species' pangenome evolution.

In their study, Joubert & Krasileva explore the (epi)genomic features that might underly such gene presence-absence variation (PAV) in two pathotypes of the fungal plant pathogen responsible for the blast disease on rice and wheat (abbreviated MoO and MoT). The major conclusion they reach is that in the two considered pathotypes, machine-learning can accurately predict which genes show PAV in each pathotype. In addition, they highlight that gene PAV associates with different genomic environments in both pathotypes.

For that purpose they first establish a map of gene PAV in genomes of rice and wheat-infecting isolates. They first show that patterns of gene PAV in rice-infecting isolates delineates known clonal lineages of the pathogen (Figure 1). Using predicted gene ontologies (GO) and PFAM annotations, they highlight that specific protein functions are more prone to PAV (Figure 2).

Next, they focus on two clonal lineages of the rice-infecting pathotype (MoO) and two of the wheat-infecting pathotype (MoT). Applying a phylogeny-informed method to call gene PAV, they conclude that the wheat-infecting pathotype (MoT) is more prone to PAV than the rice-infecting pathotype (MoO) (Figure 3). Looking at the genomic features that associate with gene PAV, they highlight contrasted genomic environments for genes showing PAV (Figure 3 and 4). Gene length and GC content in particular were found to differ between conserved and genes showing PAV (Figure 5).

To formally link genomic features to gene PAV, the authors implement a machine-learning algorithm to classify genes as conserved or displaying PAV. Major predictors of gene PAV include proximity to transposable elements, histone methylation marks and gene length together with gene expression. When trained and tested on the same pathotype, the models were able to accurately assign most of the gene set to the correct category (Figure 6). Finally, the authors end by highlighting the inaccuracy of the individual models to classify gene PAV in the alternative pathotype.

Major comments

The current study adopts an original angle to address a timely question : what drives pangenome evolution. Overall, the analysis are sound and the methods properly described, providing interesting insights into the evolutionary dynamics at place in a major plant pathogen.

My major comment is regarding to what the authors define as gene PAV along the manuscript. I would clarify what type of mechanisms might be underlying gene PAV in the current dataset. Is for instance pseudogenization considered gene PAV in the current analysis? One would expect

contrasted genomic signatures for different mechanisms leading to gene PAV (point mutations or large indels), which should be discussed in light of the machine-learning results.

Yes, this is an important point that required clarification, thank you for pointing this out. Our pipeline, specifically the TBLASTN/BLASTP steps, were implemented so that we only analyzed PAV caused by large indels (complete loss or gain of DNA), not point mutations. Specifically, our pipeline only called a missing orthogroup, if there was no DNA sequence related to that orthogroup in a genome. We have clarified this point in the Results (line 123) and the Discussion (line 380).

We also clarified in the manuscript why we analyzed large indels called using previously published Illumina sequencing data and a genome alignment-based approach. This is an orthogonal way to verify that the PAV we had identified using our pipeline were truly the result of large indels. If large indels were generating the PAV we were analyzing, then the large indels should show similar genomic features to PAV genes. This was the case, as seen in Fig S5 and S8. We have re-emphasized this point in the Results (line 244 and line 302) and the Discussion (line 420).

In addition, gene PAV are defined here at the orthogroup level, but single orthogroups can count multiple paralogs that also show PAV. I believe providing a more general view of what the pangenome looks like in terms of the proportions of core (gene present in all isolates), accessory (all but one isolate), multi-copy (paralogs) and unique (single isolate) orthogroups in the two pathotypes could help clarifying the results. In particular looking at the shape of the gene PAV accumulation curve for varying number of sampled genomes (addressing sampling bias).

Of course, as the reviewer has highlighted, PAV is a broad category that can refer to many different types of events. These events can be caused by indels (both small and large) or point mutations, and they can result in within-orthogroup CNV/PAV or PAV of whole orthogroups. In our manuscript, we have chosen to focus on whole-orthogroup PAV caused by large indels. We have chosen to focus on this specific kind of PAV because it allows us to address a specific hypothesis related to genome stability caused by DNA breaks as well as occurrences and selection of whole orthogroup losses, and test for differences in these events between rice and wheat infecting lineages. We think this is important to the evolution of *M. oryzae* as, in addition to previously published literature on PAV, we have previously observed enhanced formation of eccDNAs in *M. oryzae* (Joubert and Krasileva 2021) as well as different TE evolutionary dynamics in rice and wheat infecting lineages (Nakamoto, Joubert and Krasileva 2023) both of which can be linked to DNA breaks and genome instability. Additionally, studying within-orthogroup PAV likely would have forced us to focus on specific orthogroups of interest, such as effectors. A lot of work has been done on the expansion of effector orthogroups in fungal plant pathogens in the past (see Seong and Krasileva 2021 and 2023, for example) as well as the importance of point mutations and their role in the rapid diversification of effectors (see literature related to the two-speed genome). Therefore, we believe that our focus is more novel, and allows us to identify trends in PAV such as the fact that microbe-microbe interactions might play

an important role in *M. oryzae*'s evolution which we may have missed if we only looked at within-orthogroup PAV.

We also focused on within-orthogroup, lineage-differentiating PAV in our analysis because of the previous observations of strong phenotypic differences between the isolates of each lineage (see Thierry et al 2022). The goal of this analysis was to understand what groups or categories of genes experienced PAV between the lineages, as these groups may play a more important role than others in explaining the differences in phenotypes between these lineages. Our hypothesis was that whole-orthogroup PAV, rather than within-orthogroup CNV, was more likely to contribute to these differences in phenotypes.

Additionally, while we agree that CNV within orthogroups, especially CNV generated by tandem duplications, are also very interesting to study, within-orthogroup PAV identification would be heavily reliant on having synteny information from highly contiguous genomes that are currently not available at the pan-genome scale for *M. oryzae*. We chose instead to take advantage of the great number of available high quality short read genomes for our study. Once more highly contiguous genomes become available, pan-genome-wide within-orthogroup PAV/CNV analysis would of course be very interesting to analyze as well.

To make this deliberate choice to narrow our perspective to whole-orthogroup PAV events clearer, we have added text to in the Introduction (line 40 and 93), Results (line 119) and Discussion (line 380). We have also added Fig. S1 which is a schematic of our pipeline, which should help clarify our choice to focus on whole-orthogroup PAV events.

It is important to note that despite this choice to focus only on within-orthogroup PAV caused by losses and gains of DNA, we still saw a lot of variation in these events between MoO and MoT. We think that this category of PAV is still quite broad, and it is possible that narrowing this category down further, for example comparing large vs small indels, could be helpful in the future. Our analysis also points to the fact that comparing PAV events across species could be quite challenging to do as different PAV mechanisms will need to be taken into account in order to compare the genomic features associated with PAV across different species. We have added this point to the Discussion line 502, as well as in other areas throughout the rest of the Discussion.

Minor comments

Through the manuscript I found the reading a bit complicated sometimes. Shorter and more direct sentences could help. In addition, the back and forth between MoO / MoT, the clonal / non-clonal lineages within each pathotype and different methods to detect PAV complicate the story. Reorganising some of the paragraphs could help improving the manuscript readability.

We have made efforts to make the manuscript more concise and clarify the language throughout the manuscript to help with readability. We have also cleaned up our usage of our internally defined labels so that they are more consistent (i.e. MoO instead of "rice-infecting",

consistent usage of “reduced model” for the random forest model). We hope that this polishing helps address the reviewer’s concerns.

Introduction

I believe the introduction could benefit from a bit more context on the *Magnaporthe oryzae* species (syn. *Pyricularia oryzae*, without entering the taxonomic debate) as it has a complex history linked to plant host range / specificity and clonality / hybridization. It would set the stage when transposing the major conclusions in the context of other species (later in the discussion).

We have added more details to the Introduction regarding the *M. oryzae* species history, including the fact that there exists many host-specific pathotypes in *M. oryzae*, that isolates of each pathotype mostly form monophyletic groups, and that genetic exchange between pathotypes is thought to be rare (paragraph starting on line 72).

L9-10 > confusing

We have rephrased this question so it is hopefully clearer now (line 21).

L14 > syn. *Pyricularia oryzae*, without entering the taxonomic debate but for clarity

Of course, this is an important inclusion which we have added now.

L22 > does it mean that the prediction didn't work for the other species?

We have removed this precision.

L29 > not all microbes have expansive pangenomes (i.e closed pangenomes)

We agree and we have softened the language here (line 37).

Specify that pangenome here is considered as the set of genes, and not as the alternative definition of the set of sequences (coding and non-coding).

We believe that the following sentence makes it clear that we are discussing genes here (line 38).

L32 > not sure the reference supports the claim, evidence for HGT plus recent large transposons that associate with large genomic regions (starships).

Starships are a fascinating discovery of course. However, the idea that eukaryotes experience less HGT than prokaryotes is not a controversial one. We have, regardless, softened the language here (line 40), and added a citation to Martin 2017 which argues directly that HGT is less common in eukaryotes than prokaryotes.

L35 > pan-genome or pangenome, consistency across manuscript

We have decided to change all mention of “pan-genome” to “pangenome” throughout the manuscript.

L54-55 > facilitate how?

We have expanded our explanation of why we are interested in studying multiple populations of the same species on line 69.

L56 > No mention of the recent history of the pathogen (without focussing too much on the taxonomic debate but worth mentioning? Species complex or lineages with varying host specificity / range, potential for hybridization?

We have added more details to the *M. oryzae* species history, including the fact that there exists many host-specific pathotypes in *M. oryzae*, that isolates of each pathotype mostly form monophyletic groups, and that genetic exchange between pathotypes is thought to be rare (starting on line 72).

L68 > Related to above, clarify what is referred as "species" here, so that it is clear how it might or not relate to any other species.

We have clarified what a species represents for *M. oryzae* above.

L74 > be more specific?

We have specified the direction of all of these differences on line 97.

Results

I missed a bit of context for the analysis of structural variants based on short-reads. I would clarify how gene PAV and indels can be linked to each other (gene PAV likely generated through indels). Mentioning the proportion of indels that overlap/associate with genes could be informative for instance. Note that as presented I'm not sure one can differentiate deletions from insertions, hence indels.

L88-89 > Supplementary table with pooled metadata for the genomes could be useful (host, lineage, genome metrics, assembly ID ..)?

We agree that this information was important to include and have added it to the manuscript in the form of Table S1.

L92 > nor putative pseudogenes?

Putative pseudogenes would be counted as a presence by our pipeline, assuming that the pseudogenes resemble predicted proteins in other isolates closely enough. This is because of our thorough realignment of missing protein sequences. Again, this was a trade-off we chose in order to only count absences as sequences that were fully missing from a genome.

L98 > reference to the methods section?

We are not sure what the reviewer would like us to change here. The method we used is extensively described in the methods section, paragraph starting on line 564. Please, clarify if there is any specific information missing here.

L102 > lineage 4 seems closer to lineage 3 with effectors than with all PAV on PC1. Curious about how does a Venn diagram or upset plot looks like for all PAV and effectors only.

This is an interesting observation. A venn diagram or upsetplot would be difficult to implement for this analysis since there are so many isolates and no clear way of representing each lineage with a single data point. We calculated the euclidean distance between the centroids of the distribution of each lineage based off the first two PCs and found that lineage 4 is closer to lineage 2 (37.61) than lineage 3 (40.03) in the PCA for all orthogroups and lineage 4 is closer to lineage 3 (9.46) than lineage 2 (11.03), which confirms the reviewers analysis. We have included this analysis in our PCA code but chose not to include it in the manuscript as it is unclear what the interpretation of this result would be.

L102 > fisher test?

We have added a fisher's exact test for this comparison ($p < 0.001$, line 137).

L114 > subset for all isolates from lineage 2, 3 and 4 I presume?

Yes, only lineage 1 is omitted. We have specified this on line 148.

L119 > reference to the methods section?

We are not sure what the reviewer would like us to change here. The method we used is extensively described in the methods section, paragraph starting on line 570. Please, clarify if there is specific information missing.

Why ignore lineage specific PAV here?

We chose to focus on lineage-differentiating rather than lineage-specific PAV in our analysis because of the previous observations of strong phenotypic differences between the isolates of each lineage (see Thierry et al 2022). The goal of this analysis was to understand what groups or categories of genes experienced PAV between the lineages, as these groups may play a more important role than others in explaining the differences in phenotypes between these lineages.

We explain these differences and our reasons for focusing on them in the Results (line 111) and the Discussion (line 393).

L124-126 > Lineage-differentiating PAV orthogroups?

Yes, we have made this change (line 158).

L134 > rice-infecting here, MoO before, consistency

We have removed most mentions of rice/wheat-infecting *M. oryzae* and have replaced them with MoO/MoT for consistency. We have also added a definition of MoO and MoT to Fig. 3A to help clarify the difference between the two.

P138 > a bit confusing here what was intended and the corresponding reassignment of gene PAV across MoO lineages. Schematic could help? Or presenting the number of genes that are : monophyletic present/absent (excluded in current analysis), present/absent in at least two distinct groups, or missing in 1 genome only, could be useful to clarify the pipeline.

We agree that a schematic would be very useful to readers. We have added this schematic in the form of Fig S1. This schematic includes cartoon phylogenies representing examples of orthogroups that we called as experiencing PAV as well as examples of the ones we did not.

L154-155 > not clear why the recombining lineage would violate assumptions and was excluded

Part of our criteria for labeling a gene as experiencing PAV is that the gene must have experienced at least two independent losses. In clonal lineages, we can be fairly confident that the existence of two monophyletic groups within a lineage that are missing a gene originated from two separate gene loss events. However, in a recombining lineage, this same scenario could have arisen due to one gene loss event, and one recombination event. To avoid these scenarios, we chose to omit the recombining lineage 1. We have rephrased this sentence in the manuscript as we agree that it was not very clear which assumptions we were originally referring to (line 193). Another minor reason for excluding this lineage is that it is made up of subpopulations with very few isolates.

L165 > "in in".

We have fixed this typo. Thank you for catching it.

L168 > Supplementary table with pooled genomes metadata (host, lineage, genome metrics, assembly ID ..)?

We have added this information in the form of Table S1.

L172-173 > confusing, "datasets of comparable size for further comparisons"

We have changed this wording (line 209).

L177-178 > Add reference to the method section?

We are not sure what the reviewer would like us to change here. The method we used is extensively described in the methods section, paragraph starting on line 587.

L176 > indels rather than deletions?

Yes, indels is corrected. We have fixed mentions of deletions to indels throughout the manuscript.

L179 > would the average number of indels per genome/isolate make the statement more compelling?

Since many of the indels are the same across isolates, and since the datasets have different sizes, we chose instead to show the total number of deletions identified for each pathotype as we believe this result is more obvious and compelling for the reader.

L183 > Label for Figure 3C missing.

We have added this label, thank you for catching this mistake.

Figure 3C and others heatmaps lack a scale.

We have added this scale to all heatmaps in the manuscript.

"Much more likely", report the statistical test in the text and figures too?

We have added interquartile range, median and mean to many figures throughout the manuscript to make the statistics more obvious and clear to the readers.

L185 > No mention so far of what is the proportion of indels that overlap/associate with genes. Would clarify the link

This is shown in Fig. S5 and described on line 244.

L186 > Not clear what defined regions are here. Larger indels promote the emergence of gene PAV clusters in MoO?

We meant that PAV occurs in regions that are more isolated from conserved genes in MoO than MoT. We have clarified this (line 226).

L193 > Figure 4A-C

We have made this change.

L192-200 > provide results from statistical tests in text and figures too?

We believe strongly that adding raw numbers and statistics to these Results sections would seriously hurt the legibility of the manuscript and make for difficult and unpleasant reading since we are reporting so many different statistics. This is why we have opted to add all of the statistics and results of the statistical tests as tables in Files S5 through S11, including many that we don't mention in the text. To address this concern, however, we have added mean, median and interquartile ranges to all violin plots in the manuscript which makes the data we present more transparent and easier to interpret for readers.

L206 > Figure S4, not clear what the y axis represents given the scale and curve shape (does not seem like gene/TE counts, rather a scaled density over genomic windows?)

The y-axis does represent gene and TE counts. The x-axis is what is difficult to interpret here since it is the average counts per window across all deletion regions. Please see our explanation below.

Not clear why it caps at 1 for genes but goes over 1.5 for TEs?). The methods section mention 10 bp windows, implying that for most bins coverage values are binary (either 0 or 1).

This is partially due to the fact that RepeatMasker can generate overlapping TE annotations but FunGAP does not generate overlapping gene annotations. It may also be due to the fact that regions less than 1kbp get compressed to 1kbp so some windows can have more than 1 element because of that process.

Showing smoothed estimates (+ standard error) over longer genomic windows could provide a more direct representation of the genomic context for indels. Briefly describing (in the methods?) the scaling method (deeptools) used here could help clarify the figure.

This is an average of all the deletion regions, scaled to a length of 1kbp, along with 5kbp on either side. Using longer genomic windows here would not give a clearer picture of the results, in fact it would do the opposite as it would hide the strong peak in TEs that is right next to the deletion breakpoints, which is a key result.

Many readers that work with multi-omics data will be familiar with this type of plot since DeepTools is a standard tool in the field. We have provided an explanation of the method used in the Methods section for clarity (line 661).

Adding legend to the plot itself (next to coloured lines) would facilitate the reading I think.

We have added this legend.

L215-216 > "and some differences in these features" reads vague

We agree and we have shortened the title for this section.

L236-237 > Figure 5 (and supplementary), the density plots make differences hard to visualize. Violin plots as an alternative? Add statistical annotation to the figure itself?

We have replaced all density plots with violin plots. This is a great suggestion and it makes the interpretation much clearer. We have also added interquartile range rectangles, as well as lines for the medians and dot for the means to these plots.

L242 > Unclear what was done here

This is explained in more detail in the Methods section (line 645), but we have also added some text to the Results to clarify (line 281).

L250 > unclear what "tighter signal distribution" means here

We agree that this was too vague. We directly mention a smaller standard deviation and interquartile range instead (line 292).

L262-263 > above it is stated that "average expression was higher both in culture and in planta for conserved genes than for PAV genes", correct?

Yes, this was a mistake, which we have corrected now. We have rewritten the paragraph to point to the fact that the majority of the trends we observed in PAV genes were also observed in genomic deletions which matches the fact that our pipeline was focused on identifying PAV events caused by indels (line 302).

L266 > Phrasing. "Genomic and epigenomic features can be used to generate predictive models of gene presence-absence variation in rice and wheat-infecting *M. oryzae*"?

We agree that this sounds clearer and have implemented this change.

L278-281 > long sentence that reads more like discussion. Confusing what the values in Figure S9 represent (decrease in F1 or R2 metrics, so that the higher the value the stronger the predictor?)

We have moved these sentences to the discussion (line 413) and have rephrased the original sentences to be more clearly results (line 323). We also added a sentence to the legend of Fig. S10 to clarify the interpretation of these values. And yes, a greater decrease in F1 or R2 statistics indicate that these variables are stronger predictors of the corresponding variable.

L284 > clarify what is meant by reduced model (not always consistent in text and figures). Some variables such as epigenetic marks not available for MoT?

Yes, some variables are not available for MoT. We mentioned this in the legend for Fig. 5 and Fig. S6, but we have now added it a few other places within the manuscript to make this clearer. Specifically, on line 280 where we first introduce these datasets, and on line 327 when we first introduce the MoT model. We then cleaned up our use of "reduced MoO model", by clearly naming it "reduced MoO model" on line 332 when it is introduced, and by mentioning the "reduced MoO model" for each panel where data for that model is shown in Fig. 6.

L293 > specify which differences

We have reiterated the relevant results on line 339.

L318-320 > Providing raw numbers of true/false positives/negatives in confusion matrices would help clarify the point (Figure 6 and Figure S10)

We have added these raw counts in the form of Fig. S11.

L323 > recapitulate which are the patterns

We have added a summary of these patterns on line 370.

Discussion

As for the introduction, I missed some information related to the evolutionary history of *M. oryzae* as a fungal pathogen.

We have added this information to the introduction paragraph starting on line 72.

In addition, the authors mention strong differences in TE content between pathotypes, any explanations in light of their results (i.e MoO bears more TEs but experience less PAV compared to MoT)?

We mention this point in the results section when we start to look at other features in the genome besides TEs (line 260). We have also added a line in the discussion section to reiterate this point when we discuss the lack of evidence for a causal relationship between TEs and rapid evolution in *M. oryzae* (line 433). Our conclusion regarding this observation and our results is that, while TEs appear to be a marker for rapid evolution, they do not seem to be the whole story.

Additionally, we have investigated these TE dynamics in depth in another study, see Nakamoto, Joubert and Krasileva, 2023, which we have cited in this manuscript.

L389 > these could well be true positives not detected given the current sampling.

We agree, this is an important point to bring up in the discussion, we have added it to line 460.

Methods

Methods are well presented and all the data generated in the manuscript together with the associated scripts are fully available online, I really appreciate it.

L457-463 > what was the set size for this preliminary fine-tuning step?

To do this, we re-ran our orthogrouping approach without a single isolate's proteome (CH0043) and then used this orthogrouping to optimize our parameters. We have greatly expanded our description of the method to make it more specific and reproducible (line 552).

L464 > intermediary step to convert tblastn output to gff, additional AGAT options?

This was done using custom scripts, which we have now specified in the Methods section (line 542). The script in question is available here:

https://github.com/pierrj/moryzae_pav_manuscript_code/blob/main/validate_missing_orthogroups/parse_tblastn_hits.py

No additional AGAT options were used aside from standard input/output arguments.

L467 > not clear to me

We have clarified this to specify that we used the most common orthogroup among the 100 best matching sequences to identify which orthogroup our missing sequence would have belonged to (line 645).

L564 > Not clear what "profile plot" refers to. "Window-based coverage calculation of genes and transposons"?

This was jargon taken from the DeepTools suite of tools. We agree that isn't very clear so we changed the title to "Window-based density plots of gene and TE content for genomic regions" and removed all mention of "profile plots".

L579-580 > Got confused by the phrasing, all genomes from lineages 2 and 3 except 2 times 4 genomes were used as the training set and the 8 excluded genomes were then used for testing, correct?

This is correct. There was a typo (testing -> training) which we have fixed, and specified on line 671 that the 2 times 4 genomes are saved for training.

Reviewer #2 (Comments for the Authors (Required)):

In this study, the authors look at gene presence-absence polymorphism in different pathotypes of an important fungal plant pathogen using publicly available genomic data.

I have three major comments:

1) the authors didn't mention whether they consider or not the dispensable chromosomes previously described in the studied species; substantial gene presence-absence polymorphism is expected on dispensable chromosomes and comparison between core and dispensable chromosomes could be interesting ; if the author focus on core chromosomes only it should be explained to interpret this study

Isolation and identification of dispensable chromosomes is technically challenging and requires extensive isolation and sequencing work so it is unfortunately beyond the scope of this study (see Langner et al. 2021). While dispensable chromosomes in *M. oryzae* reflect some of the features we analyze in this study (high TE density, low gene density), they are thought to be made-up of chromosomal sequences and generated through large structural rearrangements. Dispensable chromosomes in *M. oryzae* are not as common as other fungi (such as *Zymoseptoria trici* or *Fusarium oxysporum*), and do not contain drastically different sets of genes. This is why we have chosen not to discuss them in the context of *M. oryzae*. Again, since dispensable chromosomes contain chromosomal sequences in *M. oryzae* this should not have impacted our PAV analysis. This is a good point for future study.

2) I didn't understand well the gene enrichment analyses results; the HET gene and genes related to antibiotic production didn't appear much more enriched than other gene categories ; in any case, I found the discussion about microbiome importance very speculative as finding genes related to antibiotic production doesn't imply necessarily a link with antibiotic production; if it is the case, references should be provided

Only GO terms and domains that were assigned to three or more lineage-differentiating PAV orthogroups and with enrichment p-values less than 0.05 were reported in Fig. 2. This means that all categories shown in the figure represented a statistically significant enrichment. All other categories were not reported. We have added this information to the legend to Fig. 2. We chose to focus on this result because multiple GO terms and PFAM domains pointed towards non-self interactions, especially since it was both antibiotic production (inhibition of bacteria outside of the fungus) and non-self recognition (prevention of bacterial infection). We have re-emphasized this point in the discussion (line 390).

Another reason we emphasized this result is because studies of plant pathogens often focus only on how pathogens interact with their plant hosts and ignore how the pathogens adapt to their environments, including other microbes. Our result hints that adaptation to interactions with other microbes may play a role in local adaptation in this pathogen. We emphasize this point in our discussion (line 400).

Additionally, it is important to note that we are not hoping to discover new antimicrobials through this analysis. We are simply showing evidence that they play an important role in the evolution of *M. oryzae*.

Finally, this interpretation of course relies on different landraces of rice plants having different microbiomes. We have added a few citations to the discussion that demonstrates this phenomenon (line 395).

3)7) information on quality of genome assemblies (N50, L50, etc..) and genome coverage of Illumina reads used for deletion detection should be given to assess the power of detecting PAV in all strains; in particular differences described between the pathotypes might be due to technical aspects only

We have provided a table of genome quality metrics for all assemblies, Table S1. PAV genes were not called using deletion detection in our study, they were only called using genome assemblies. We used an ensemble approach for deletion detection in our study that uses many different deletion calling software (line 592). Each software has different requirements for read coverage for calling deletions. We provided the parameters used for each software in that methods paragraph.

We do not believe that differences in genome quality affected our comparisons between the two pathotypes. Based on the statistics shown in Table S1, the genome qualities we used for MoO and MoT were very similar, especially given the limited set of available genomes. Additionally, we analyzed and displayed our results in many different ways so as to rule out any effects of

differences in genome quality. See for example Fig. 3 C-E. While the results in panel C could be affected by genome quality because we are measuring distance to the nearest feature, the results in panel E are much more robust to this issue since we are only looking at the flanking 1000 bp regions. We have added a sentence to clarify that this analysis is intended to verify whether the results are robust to differences in genome quality (line 222).

I have the following minor comments:

1) the introduction could be reframed as the authors highlight some limits of previous studies which in my opinion were actually addressed in these studies; see reference 6 as an example; the authors can present their study with other arguments

We agree that the original phrasing did not fully reflect the literature and have re-phrased this sentence to show that we are building off of this previous work and applying it to a new system (line 49). Furthermore, we have significantly re-written the first paragraph of the introduction to have a more positive voice and framing to address this concern.

2) statistical testing is lacking for several analyses (ex: L103-106)

We have performed a Fisher's exact test for this comparison and have added the result to line 136 ($p < 0.001$).

3) L147: it is unclear what is a subclade in a lineage and how the author define it

We explicitly defined what we meant by subclades on line 182. We also provided a new supplemental figure (Fig. S1) showing examples of what we mean here.

4) L310 : subtitle is too long

We have removed the last clause to shorten the subtitle.

5) L370 : it is unclear what are subterminal regions and what link did the author found

We referred to subterminal regions as "chromosome ends" in the Introduction section (line 47). We also clarified in the Discussion that the association of PAV genes to TEs is the only link to subterminal regions that we were able to find in our analysis (line 437). This is because of issues with genome quality, as explained in the previous sentence.

6) L388 : "many genes" is not informative

We clarified by providing the percentage of genes that were labeled "PAV" by the RF classifier when the true label was supposed to be "not PAV". See line 459.

September 19, 2023

GENETICS-2023-306426

Distinct genomic contexts predict gene presence-absence variation in different pathotypes of *Magnaporthe oryzae*

Dear Dr. Krasileva:

Two experts in the field have reviewed your manuscript, and I have read it as well. I am pleased to inform you that, with minor revisions, it is potentially suitable for publication in GENETICS. The reviewers have comments and concerns that need to be addressed in a revised manuscript. You can read their reviews at the end of this email.

It is most important that you address the following in your resubmission:

- the reviewer #3 concerns, which bring up some additional points not previously raised and the concern about why wheat and rice lineages are treated differently need to be addressed
- the reviewer has asked about the corrected counts for lineages in the tables and other places
- there are additional points raised by the reviewer about how the removal of the MoT lineages changes the interpretations - can you provide some response or adjustment to the text to better understand if anything needs to be restated in light of this.

We look forward to receiving your revised manuscript. Please let the editorial office know approximately how long you expect to need for revisions.

Upon resubmission, please include:

1. A clean version of your manuscript;
2. A marked version of your manuscript in which you highlight significant revisions carried out in response to the major points raised by the editor/reviewers (track changes is acceptable if preferred);
3. A detailed response to the editor's/reviewers' comments and to the concerns listed above. Please reference line numbers in this response to aid the editors.

Additionally, please ensure that your resubmission is formatted for GENETICS.

<https://academic.oup.com/genetics/pages/general-instructions>

Follow this link to submit the revised manuscript: Link Not Available

Sincerely,

Jason Stajich
Associate Editor
GENETICS

Approved by:
David Begun
Senior Editor
GENETICS

Reviewer #1 (Comments for the Authors (Required)):

I believe that this revised version of the manuscript well addressed most of the initial comments, congratulation to the authors.

As minor comment, I think it would be valuable to provide the number of orthogroups identified in each pathotype / lineage and their PAV status in regard to the number of genes involved (single copy / multi copy orthogroup) as it has direct implications on the putative mechanisms underlying PAV.

I would also provide the statistics for the violin plots presented across multiple figures.

L. 67 > "to occur to occur", "to locate" instead?

L. 107 > not sure one can differentiate between loss or gain at this stage

L. 222 > providing the average number of intel per isolate / genome would be more informative?

- L. 276 > Figure 5E labelled as "Rice" (MoO)
- L. 288-289 > not sure why "transfer" to other strains? Is it not based on a reference assembly?
- L. 388 > "rather than pseudogenisation" ?

Reviewer #3 (Comments for the Authors (Required)):

In this paper Joubert and Krasileva present an innovative way to address an important question in plant pathogen evolution: How does PAV shape the pangenome evolution? The authors then show that machine-learning can accurately predict which genes are likely to show PAV.

The article has been reviewed before by two reviewers. I took their comments and the authors response into consideration, but also added a few points that I think should be addressed before publication.

In their response to the editor, the authors also mention a mistake in the previous version, where some genome assemblies were mislabelled as MoT. These assemblies, which represented an entire lineage (one of two MoT lineages), have been removed in the current version.

Overall, the manuscript is well-written and provides interesting conclusions. However, I think some more quality checks are warranted given that the dataset changed quite drastically from the last version.

I have following major comments:

- Regarding the comment of previous reviewers: Could genome quality influence the analysis?

The authors state that there is no noticeable difference in genome quality. However, I'm not sure if the provided data suffices to address the reviewers comments. Table S1 shows that there are some "outlier" assemblies from this lineage which are assembled to chromosome quality (or quasi Chr quality). These assemblies seem to have a strong influence on the average N50 of the analysed lineages (AVR N50 rice infecting lineage = 133 kb; AVR N50 wheat infecting lineage = 517 kb). Could these differences contribute to the gene distance analysis? Providing more detailed data which was used to generate Fig 3 and S4 (e.g., Average gene distance per assembly) might clarify if the assembly quality influences these analyses. Given the drastically reduced sample number in the wheat infecting lineage compared to the previous version of the manuscript, more quality control of the dataset would strengthen the conclusions of this paper. The (somewhat) bimodal distribution of Indel length in Fig 3B might indicate an effect of high quality assemblies that are only present in the wheat infecting lineage.

Mapping the location of PAV genes on some of the high quality assemblies of the rice infecting lineage in comparison with contig breaks in short read assemblies might also provide useful insight into the quality of the gene distance analysis, as it is expected that repeat content, PAV, and contig breaks coincide to some extent in the short read assemblies (based on previous studies). Together these factors might influence the outcome of the presented analyses. I think especially as the set of genome assemblies significantly changed from previous version, more quality control on the dataset is warranted.

-This was a minor comment in the previous version, but in my opinion this is actually a major concern in the current experimental setup and the response to previous comments is in my view insufficient, especially given that 2/3 of the MoT genomes have been removed, which only left 1 lineage and this lineage likely shows signatures of recombination (based on previous studies) seem to be more extensive than in the recombining MoO lineage (see e.g. Gladioux et al., 2018).

"Reviewer comment: L154-155 > not clear why the recombining lineage would violate assumptions and was excluded

Author response: Part of our criteria for labeling a gene as experiencing PAV is that the gene must have experienced at least two independent losses. In clonal lineages, we can be fairly confident that the existence of two monophyletic groups within a lineage that are missing a gene originated from two separate gene loss events. However, in a recombining lineage, this same scenario could have arisen due to one gene loss event, and one recombination event. To avoid these scenarios, we chose to omit the recombining lineage 1. We have rephrased this sentence in the manuscript as we agree that it was not very clear which assumptions we were originally referring to (line 193). Another minor reason for excluding this lineage is that it is made up of subpopulations with very few isolates"

My concern is, that the authors exclude some rice infecting lineages from the analysis, either because of sample numbers or due to signatures of recombination. However, previous studies have also shown similar signatures of admixture/recombination in

the wheat infecting lineage. The criteria to include/exclude lineages are not clear and might have a major impact on the outcome of the analysis.

In the same context, the sentence in line 215 and 216 might be a bit misleading, as recent studies have at least identified one clonal lineage (the B71 lineage, Latorre et al., 2023 and Liu et al., 2022) and while the cited reference (Rahnama et al., 2021) suggests a clonal propagation of the wheat infecting lineage in the very recent history, they also suggest extensive recombination events between the 1950s and 1990s that shaped the entire lineage.

In conclusion, I do not fully understand why the authors treat the wheat and rice infecting lineage differently (I think at least repeating the analysis on the entire rice infecting lineage, using the same parameters as for the wheat infecting lineage would be necessary to exclude a sampling bias in the analysis. This could significantly affect the conclusion that PAV are 50% more common in MoT.

Minor:

- Adding some available metadata to the assembly names could also help with the interpretation of this study. Specifically, in the Magnaporthe community, it is common to add the isolate ID instead of the assembly ID. Such information would make the interpretation of the data much easier for the reader. In table S1 the authors show isolate IDs for MoO but assembly ID for MoT. This should be more consistent.

- Could the authors also provide a more detailed summary of the total number of orthogroups identified in the study? Especially for Fig 1 it is not clear how many orthogroups there are in total. This would help to put the number of lineage defining or effector orthogroups into context.

- It would help to visualize the overlap of "lineage defining orthogroups" and "effector orthogroups" in figure 1. Just from the text it is not obvious how these two groups relate to each other.

- Could the authors give a bit more detail how the thresholds for "lineage differentiating orthogroups" were chosen (line 136: 587 orthogroups that represent 70.53% of PC1 and 62.17% of PC2)? In the Methods section they write "orthogroups that contributed more than 0.1% of the variance. Could the authors please clarify.

- Line 169: non-host recognition should probably be non-self recognition

- As one of the previous reviewers already pointed out, the GO term and PFAM enrichment analysis seems a bit undifferentiated. The strong focus on HET/NLR genes seems unwarranted given the results of this analysis. This analysis would also provide a good opportunity to link the observed PAV orthogroups to previously published results: E.g., does the group secondary metabolites and KR / Dehydratase (polyketide synthesis) include the known secondary metabolite cluster including ACE1 that has known avirulence activity? In addition, metalloproteases come up in this analysis which might relate to the known avirulence effector AVR-Pita, that is known to undergo frequent rearrangements (see e.g. Chuma et al., 2011).

- Is the difference in Indel length presented in figure 3B statistically significant? The authors only provide details for panels C-E but not for B.

- In line 290 the authors state that no expression data was available for MoO, but Fig 5 shows expression values for MoO. Also, in the methods section the authors describe a source of expression data for MoO (Zhang et al., 2021) Could the authors please clarify?

- Is it possible that there are different driving forces that generate PAV in recombining and clonal lineages? This point is quite vaguely discussed towards the end of the discussion ("different evolutionary histories"). The authors excluded the potentially recombining rice infecting lineage from the analysis and only focussed on two clonal lineages, while the wheat infecting lineage shows similar pattern of recombination. Meiotic and somatic recombination or hybridization events could lead to different PAV pattern. This could be tested using the recombining rice lineage in comparison to the wheat infecting lineage. Such a test could provide insights into several of the discussion points the authors raise (differences in host adapted lineages, possible connection to microbiome, contribution of genomic features...).

- Line 489 "9,800 thousand years ago" the "thousand" should be deleted

- The sentence in lines 38-43 could be simplified to make it clearer and easier to read.

- In the response to the editor the authors explain that 52 out of the 88 MoT genomes were mislabelled and removed in the current version. That would leave 36 MoT genomes that are also present in Table S1. However, in the Methods they talk about 47 datasets for MoT. Could the authors please clarify?

Associate Editor Comments:

Two experts in the field have reviewed your manuscript, and I have read it as well. I am pleased to inform you that, with minor revisions, it is potentially suitable for publication in GENETICS. The reviewers have comments and concerns that need to be addressed in a revised manuscript. You can read their reviews at the end of this email.

It is most important that you address the following in your resubmission:

- the reviewer #3 concerns, which bring up some additional points not previously raised and the concern about why wheat and rice lineages are treated differently need to be addressed
- the reviewer has asked about the corrected counts for lineages in the tables and other places
- there are additional points raised by the reviewer about how the removal of the MoT lineages changes the interpretations - can you provide some response or adjustment to the text to better understand if anything needs to be restated in light of this.

We have incorporated the reviewers' feedback into a new draft of the manuscript. We have made small changes to the main text to clarify some points raised by the reviewers and have added new statistics to address reviewers' comments in Files S5, S6 and S8. We have updated the names of the MoT isolates in our phylogeny in Fig S3 and Table S1 to reflect strain names instead of assembly names. We have added a mention in the Results and Discussion of the pandemic clonal lineage of MoT as brought to our attention by reviewer #3. We have also included the number of PAV orthogroups for this lineage alone as suggested by the reviewer. Finally, we have provided detailed clarifications and responses to each of the reviewers' comments.

Reviewer #1 (Comments for the Authors (Required)):

I believe that this revised version of the manuscript well addressed most of the initial comments, congratulation to the authors.

As minor comment, I think it would be valuable to provide the number of orthogroups identified in each pathotype / lineage and their PAV status in regard to the number of genes involved (single copy / multi copy orthogroup) as it has direct implications on the putative mechanisms underlying PAV.

The number of orthogroups identified in each pathotype/lineage and their PAV status are shown in Fig. 3A. As described throughout the manuscript, we did not differentiate between single copy and multi-copy orthogroups and focused instead on orthogroups that were either present (in one or multiple copies) or entirely absent from each genome to make our conclusions robust. We believe that at this point, correct identification and counting copy number variation while interesting to us as well requires long read assemblies and types of annotation not available yet for Magnaporthe, therefore this type of information would be beyond the scope of this manuscript.

I would also provide the statistics for the violin plots presented across multiple figures.

Thank you for pointing this out. The majority were already included in Files S6 through S12 but we have added all statistics from the violin plots to Files S5, S6 and S8 now.

L. 67 > "to occur to occur", "to locate" instead?

We've fixed this typo.

L. 107 > not sure one can differentiate between loss or gain at this stage

Thank you, we fixed the majority of these mistakes in the first round of reviews but this was one we missed.

L. 222 > providing the average number of indel per isolate / genome would be more informative?

We believe that the numbers we reported (total number of identified indel for each pathotype) despite the great difference in the number of datasets (double for MoO than MoT) provide clear evidence of the point that we are making in this section - that large indels are much more common in MoT than MoO. Because of the complications introduced by merging indel breakpoints across samples, as well as the reference-based nature of indel calling and the heterogeneous distribution of these indels across samples, we have chosen not to complicate our message by introducing more complex summary statistics here.

L. 276 > Figure 5E labelled as "Rice" (MoO)

Thank you for catching this mistake. We have fixed it now.

L. 288-289 > not sure why "transfer" to other strains? Is it not based on a reference assembly?

All of the next-generation sequence data we obtained was mapped to the reference strain, and then the values for each orthogroup in the reference strain were applied to the other assemblies. This was done so that we could get per-gene values for each next-generation sequencing dataset for all of the genes across all of the genomes we used in the manuscript. We have detailed the methods we used for this in the "Next-generation sequencing data and GC content analysis" section of the Methods and have detailed the potential caveats and pitfalls of this approach in the Discussion section.

L. 388 > "rather than pseudogenisation" ?

We've added the mention of pseudogenisation along with gene silencing in this sentence.

Reviewer #3 (Comments for the Authors (Required)):

In this paper Joubert and Krasileva present an innovative way to address an important question in plant pathogen evolution: How does PAV shape the pangenome evolution? The authors then show that machine-learning can accurately predict which genes are likely to show PAV.

The article has been reviewed before by two reviewers. I took their comments and the authors response into consideration, but also added a few points that I think should be addressed before publication.

In their response to the editor, the authors also mention a mistake in the previous version, where some genome assemblies were mislabelled as MoT. These assemblies, which represented an entire lineage (one of two MoT lineages), have been removed in the current version.

Overall, the manuscript is well-written and provides interesting conclusions. However, I think some more quality checks are warranted given that the dataset changed quite drastically from the last version.

I have following major comments:

- Regarding the comment of previous reviewers: Could genome quality influence the analysis?

The authors state that there is no noticeable difference in genome quality. However, I'm not sure if the provided data suffices to address the reviewers comments. Table S1 shows that there are some "outlier" assemblies from this lineage which are assembled to chromosome quality (or quasi Chr quality). These assemblies seem to have a strong influence on the average N50 of the analysed lineages (AVR N50 rice infecting lineage = 133 kb; AVR N50 wheat infecting lineage = 517 kb). Could these differences contribute to the gene distance analysis? Providing more detailed data which was used to generate Fig 3 and S4 (e.g., Average gene distance per assembly) might clarify if the assembly quality influences these analyses.

We agree that there are differences in genome quality between the MoT and MoO dataset and do not argue otherwise. We observed that the MoT genomes were not of lower quality and especially BUSCO completeness than the MoO genomes since we did quality checks to ensure that that was the case. It is true that a few of these genomes are very high quality and that is a possibility that these could affect the analysis. We have now made this more explicit when we first introduce these genomes on line 213 of the Results section.

However, only a handful of assemblies were of chromosome or near-chromosome level quality and since we reported averages for all genes in these figures, we do not believe that these assemblies would have had a huge impact on our results. We also focused on analyses that should not be affected by genome quality which we believe support our conclusions. The first is

our use of a cutoff and qualitative binning of genes to see how many PAV vs conserved genes are close to PAV genes/TEs/genes (see Fig 3E, Fig 4C and Fig S4C.) We specifically mention the goal of addressing differences in genome quality when we first introduce these analyses (line 235). All of these analyses resulted in similar results to our distance-based analyses which showed that genome quality did not have much of an impact on the conclusions drawn from these results. We also included Fig 3D, Fig 4B and Fig S4B, all of which only relied on the genes having a nearest element in one direction rather than both (like Fig 3C, Fig 4A and Fig S4B) which were not as strongly affected by genome quality as the 2D-density plots. Again these results supported the general conclusions we drew in the manuscript.

We have also included an analysis of indels called using Illumina sequencing data and high-quality reference genomes for both MoO (Guy-11 reference assembly by Bao et al 2017, which we did not use as an isolate in our PAV analysis) and MoT (B71 reference assembly, also used as part of the dataset for the rest of the study) in Fig3B, Fig S5, and Fig S8. Since these analyses were performed using only a high quality reference and short-read sequencing data for both pathotypes, and no other assemblies, these analyses were not affected by assembly quality. We believe that our finding that indels in MoT tend to be smaller than those in MoO (Fig 3B) support our findings that PAV is more likely to occur in clusters in MoO than in MoT as shown in Fig. 3C-E and, again, this finding is not affected by genome quality.

Additionally, through these analyses, we were able to confirm the results we observed in PAV vs conserved genes for both the next-generation sequencing datasets we analyzed (Fig S8) as well as for the distance to genes and TEs (Fig. S5) which was the result that was most likely to be affected by genome quality issues. Though these results do not directly relate to the differences in genome qualities between the MoO and MoT datasets, these analyses show that the results we present are generally robust to the pitfalls of using low quality genome assemblies.

Finally, of course it is true that a more homogenous dataset would be ideal here. However, at the time that the study was conducted, we believe that we used the best possible dataset that we had access to and we focused on the type of analyses that would be robust given the data such as looking at whole orthogroup loss rather than individual gene copy number variation within orthogroups.

Given the drastically reduced sample number in the wheat infecting lineage compared to the previous version of the manuscript, more quality control of the dataset would strengthen the conclusions of this paper. The (somewhat) bimodal distribution of Indel length in Fig 3B might indicate an effect of high quality assemblies that are only present in the wheat infecting lineage.

Again, we only used a high-quality reference and short read sequencing data to call indels in MoO and MoT for Fig. 3B. We did not use any other genome assemblies to generate this data and therefore it was not affected by differences in genome quality between MoO and MoT.

This bimodal distribution is quite interesting, however. It could indicate that different types of PAV are occurring in MoT with different mechanisms and could be a promising topic for followup studies. As described above, ensuring orthogonal approaches that are independent of reference quality helped us to support our conclusions.

Mapping the location of PAV genes on some of the high quality assemblies of the rice infecting lineage in comparison with contig breaks in short read assemblies might also provide useful insight into the quality of the gene distance analysis, as it is expected that repeat content, PAV, and contig breaks coincide to some extent in the short read assemblies (based on previous studies). Together these factors might influence the outcome of the presented analyses. I think especially as the set of genome assemblies significantly changed from previous version, more quality control on the dataset is warranted.

We agree with the reviewer, it is clear that contig breaks and repeat content tend to be associated which is why we included the previously mentioned analyses to make sure that the conclusions we drew from our results were not affected by this association.

-This was a minor comment in the previous version, but in my opinion this is actually a major concern in the current experimental setup and the response to previous comments is in my view insufficient, especially given that 2/3 of the MoT genomes have been removed, which only left 1 lineage and this lineage likely shows signatures of recombination (based on previous studies) seem to be more extensive than in the recombining MoO lineage (see e.g. Gladieux et al., 2018).

"Reviewer comment: L154-155 > not clear why the recombining lineage would violate assumptions and was excluded

Author response: Part of our criteria for labeling a gene as experiencing PAV is that the gene must have experienced at least two independent losses. In clonal lineages, we can be fairly confident that

the existence of two monophyletic groups within a lineage that are missing a gene originated from two separate gene loss events. However, in a recombining lineage, this same scenario could have arisen due to one gene loss event, and one recombination event. To avoid these scenarios, we chose to omit the recombining lineage 1. We have rephrased this sentence in the manuscript as we agree that it was not very clear which assumptions we were originally referring to (line 193). Another minor reason for excluding this lineage is that it is made up of subpopulations with very few isolates"

My concern is, that the authors exclude some rice infecting lineages from the analysis, either because of sample numbers or due to signatures of recombination. However, previous studies have also shown similar signatures of admixture/recombination in the wheat infecting lineage. The criteria to include/exclude lineages are not clear and might have a major impact on the outcome of the analysis.

In the same context, the sentence in line 215 and 216 might be a bit misleading, as recent studies have at least identified one clonal lineage (the B71 lineage, Latorre et al., 2023 and Liu

et al., 2022) and while the cited reference (Rahnama et al., 2021) suggests a clonal propagation of the wheat infecting lineage in the very recent history, they also suggest extensive recombination events between the 1950s and 1990s that shaped the entire lineage. In conclusion, I do not fully understand why the authors treat the wheat and rice infecting lineage differently (I think at least repeating the analysis on the entire rice infecting lineage, using the same parameters as for the wheat infecting lineage would be necessary to exclude a sampling bias in the analysis. This could significantly affect the conclusion that PAV are 50% more common in MoT.

Thank you for identifying the information we missed from Latorre et al. 2023 regarding the pandemic clonal lineage of MoT. It is clear that this lineage appears as a sub-clade in our phylogeny of MoT isolates shown in Fig. S3 (now marked by asterisks). To address the reviewer's point, we repeated our PAV orthogroup counting with just this subclade and have included the results of the analysis in the text (line 222). This re-analysis identified only 789 PAV orthogroups for this sub-lineage, which is fewer than either MoO lineage. However, we point out in the text that this sub-lineage is far smaller than either MoO lineage (24 isolates in the B71 MoT lineage, vs 32 and 48 for the MoO lineages) which may have had an impact on the result.

We would also like to re-emphasize the result of our analysis of indels in MoO and MoT (Fig. 3B) which showed similar numbers of indels in MoO and MoT despite using far fewer Illumina sequencing datasets for MoO (117) than MoT (47). This result supports our hypothesis that PAV is more common in MoT than MoO. Additionally, the Illumina sequencing datasets we used for this analysis were not segregated by lineage and therefore they do not have the same issues of uneven treatment or potential arbitrary ruling out of certain lineages between MoO and MoT that the reviewer mentions. Therefore, we believe that the results of these analyses are robust.

We understand that the difference in sequence diversity and evolutionary time between our MoO and MoT datasets is a significant caveat to this analysis. We mention it in the Discussion on line 487 and now specifically mention the result of the fewer number of PAV orthogroups in the pandemic lineage. However, the ideal dataset which includes an equal number of MoO and MoT genomes equally spaced apart in evolutionary time does not currently exist. Therefore, we believe we did the best possible analysis that we could given the data we had access to during this study.

We stand by the claim that the MoT lineage reproduces primarily clonally as it is a part of the core hypothesis of Rahnama et al 2023 (the preprint was previously cited in the text). Their data shows that the MoT lineage was originally generated by two rounds of sexual recombination followed by subsequent clonal reproduction and selection by the host. Therefore, the genomes we use generally satisfy the assumptions of no sexual recombination that were necessary for the method we used to identify PAV orthogroups.

Finally, the criteria that we use to include or exclude lineages of MoO and MoT are clearly described on line 194. For MoO, we excluded lineage 4 because it included too few isolates (7) and we excluded lineage 1 because it is sexually recombining which violates the assumption of

clonality of our PAV orthogroup calling method (Fig. S1). We also point out on line 227 that the analysis of indels that we performed included Illumina sequencing data from all MoO lineages and supported our findings for lineage 2 and 3 for MoO.

Minor:

- Adding some available metadata to the assembly names could also help with the interpretation of this study. Specifically, in the Magnaporthe community, it is common to add the isolate ID instead of the assembly ID. Such information would make the interpretation of the data much easier for the reader. In table S1 the authors show isolate IDs for MoO but assembly ID for MoT. This should be more consistent.

We have added the MoT strain names to table S1 and to the MoT phylogeny in Fig. S3.

- Could the authors also provide a more detailed summary of the total number of orthogroups identified in the study? Especially for Fig 1 it is not clear how many orthogroups there are in total. This would help to put the number of lineage defining or effector orthogroups into context.

We have added the total number of orthogroups to the first paragraph of the Results section, where we initially described our pipeline. This information is also shown in Fig. 3A.

- It would help to visualize the overlap of "lineage defining orthogroups" and "effector orthogroups" in figure 1. Just from the text it is not obvious how these two groups relate to each other.

In Fig. 1C, only lineage-differentiating PAV orthogroups are represented. We previously omitted to specifically state this in the figure legend but we have added it now. We have also added the title "Lineage-differentiating PAV Orthogroups" at the top of Fig. 1C to make this clearer to the readers.

- Could the authors give a bit more detail how the thresholds for "lineage differentiating orthogroups" were chosen (line 136: 587 orthogroups that represent 70.53% of PC1 and 62.17% of PC2)? In the Methods section they write "orthogroups that contributed more than 0.1% of the variance. Could the authors please clarify.

The `get_pca_var` function in R was used to obtain the amount that each orthogroup contributed to the variance of PCs 1 and 2. Orthogroups that contributed more than 0.1% of this variance in either PC1 or PC2 were labeled lineage-differentiating PAV orthogroups. In total, these orthogroups explained 70.53% of the variance in PC1 and 62.17% of the variance in PC2.

We have reworded the Methods paragraph to clarify this point. We have also removed the percentages from the Results section and moved them to the methods section for clarity.

- Line 169: non-host recognition should probably be non-self recognition

Thank you for catching this mistake. We have fixed it now.

- As one of the previous reviewers already pointed out, the GO term and PFAM enrichment analysis seems a bit undifferentiated. The strong focus on HET/NLR genes seems unwarranted given the results of this analysis. This analysis would also provide a good opportunity to link the observed PAV orthogroups to previously published results: E.g., does the group secondary metabolites and KR / Dehydratase (polyketide synthesis) include the known secondary metabolite cluster including ACE1 that has known avirulence activity? In addition, metalloproteases come up in this analysis which might relate to the known avirulence effector AVR-Pita, that is known to undergo frequent rearrangements (see e.g. Chuma et al., 2011).

As we have previously explained in our first response to reviewers, the reason we highlight antibiotic production and non-self-recognition is precisely because these results were different from what we and the field originally expected. We hold this enrichment to the same level of importance that effectors are enriched amongst lineage-differentiating PAV genes in the manuscript and do not focus much detail as functional annotation of PAV is not the main focus of this study. Our goal is to point out that we observe this enrichment and, significantly, that interactions with microbes are represented as two separate categories: antibiotic production AND non-self-recognition.

As for the specific effector analyses that the reviewer suggests, yes it is very likely that these other enriched terms are enriched precisely because effectors themselves are enriched. Since we already show that effectors are enriched amongst lineage-differentiating PAV orthogroups, we do not believe that these follow up analyses are necessary as these type of analyses had been extensively done for Magnaporthe in the past.

To clarify this point, we have also changed our reference to antibiotic production and non-self-recognition in the Abstract and Introduction to a more general “microbe-microbe interactions”.

- Is the difference in Indel length presented in figure 3B statistically significant? The authors only provide details for panels C-E but not for B.

Yes, this difference is statistically significant ($p < 0.001$ from a permutation test). We have now added these statistics in file S5 and referred to them in the text.

- In line 290 the authors state that no expression data was available for MoO, but Fig 5 shows expression values for MoO. Also, in the methods section the authors describe a source of expression data for MoO (Zhang et al., 2021) Could the authors please clarify?

This line indicated that only expression data, rather than other next-generation sequencing datasets, is available for MoT rather than the fact that expression data is available for MoT but not for MoO. We have rephrased this sentence to make it clearer.

- Is it possible that there are different driving forces that generate PAV in recombining and clonal lineages? This point is quite vaguely discussed towards the end of the discussion ("different evolutionary histories").

The authors excluded the potentially recombining rice infecting lineage from the analysis and only focussed on two clonal lineages, while the wheat infecting lineage shows similar pattern of recombination. Meiotic and somatic recombination or hybridization events could lead to different PAV pattern. This could be tested using the recombining rice lineage in comparison to the wheat infecting lineage. Such a test could provide insights into several of the discussion points the authors raise (differences in host adapted lineages, possible connection to microbiome, contribution of genomic features...).

This is a great idea, and definitely should be explored in the future. It would be very interesting to do a more thorough analysis of how the recombining lineage (and sub-lineages) of *M. oryzae* differ in their PAV patterns, both between recombining and non-recombining lineages and between recombining sub-lineages. A quick look at the data presented by Thierry et al 2022 (Fig 5C) and Latorre et al 2020 (Fig. 6C) appears to show fewer missing orthogroups from lineage 1 compared to other lineages. However, we believe that this investigation is beyond the scope of the current study. This is in large part because a deeper dive into PAV in *M. oryzae* requires more sophisticated methods than simply looking at PAV matrices (as shown in those studies). A major obstacle is categorizing orthogroups into PAV-prone vs conserved categories. We used the methods described in Fig. S1 to overcome this obstacle. As shown in this figure and described in the Methods and Results sections, we rely heavily on the assumption that no recombination is happening between *M. oryzae* lineages. This assumption would be violated by extensive recombination between the isolates. Therefore, a new method would need to be used that does not depend on this assumption and so we believe that this question is beyond the scope of the manuscript.

Additionally, as cited in the Results section (line 220), according to Rahnema et al 2023, MoT is thought to have reproduced primarily clonally, after two rounds of sexual recombination that resulted in the original wheat-infecting isolates. While this means that our assumption of no-recombination is not as strong as for MoO, we believe that the reproductive history of MoT is starkly different from MoO lineage 1 and looks much more like the clonal lineages of MoO.

- Line 489 "9,800 thousand years ago" the "thousand" should be deleted

Thank you for catching this, we have fixed this typo now.

- The sentence in lines 38-43 could be simplified to make it clearer and easier to read.

We have removed the clause defining the accessory regions of the pangenome. The sentence reads much more smoothly now.

- In the response to the editor the authors explain that 52 out of the 88 MoT genomes were mislabelled and removed in the current version. That would leave 36 MoT genomes that are

also present in Table S1. However, in the Methods they talk about 47 datasets for MoT. Could the authors please clarify?

We used 36 MoT genomes because these were previously assembled MoT genomes that passed our quality control metrics and that were available on NCBI. These genomes were used for the vast majority of analyses throughout the manuscript. We used 47 Illumina sequencing datasets because these were the sequencing datasets that were available on the SRA. These datasets were only used for calling structural variants using previously published methods using sequencing read alignment to a reference genome.

December 19, 2023

RE: GENETICS-2023-306678

Dr. Ksenia V. Krasileva
University of California Berkeley
Plant and Microbial Biology
Koshland Hall
Berkeley

Dear Dr. Krasileva:

Congratulations! We are delighted to inform you that your manuscript entitled "**Distinct genomic contexts predict gene presence-absence variation in different pathotypes of *Magnaporthe oryzae***" is acceptable for publication in GENETICS. Many thanks for submitting your research to the journal.

The revised manuscript has addressed the weaknesses which were highlighted by reviewers and the changes have improved the small areas that needed a little more attention. This revision has addressed these needs and presents a compelling and detailed article on this topic of PAV in this important plant pathogen.

To Proceed to Production:

1. Format your article according to GENETICS style, as discussed at <https://academic.oup.com/genetics/pages/general-instructions>, and upload your final files at <https://genetics.msubmit.net>.
2. Your manuscript will be published as-is (unedited-as submitted, reviewed, and accepted) at the GENETICS website as an Advanced Access article and deposited into PubMed shortly after receipt of source files and the completed license to publish. Please notify sourcefiles@thegsajournals.org if you do not wish to publish your article via Advanced Access.
3. We invite you to submit an original color figure related to your paper for consideration as cover art. Please email your submission to the editorial office or upload it with your final files. You can submit a small-sized image for evaluation, and if selected, the final image must be a TIFF file 2513px wide by 3263px high (8.375 by 10.875 inches; resolution of 600ppi). Please avoid graphs and small type.

If you have any questions or encounter any problems while uploading your accepted manuscript files, please email the editorial office at sourcefiles@thegsajournals.org.

Sincerely,

Jason Stajich
Associate Editor
GENETICS

Approved by:
David Begun
Senior Editor
GENETICS

note: Please add jnls.author.support@oup.com and genetics.oup@kwgglobal.com (or the domains @oup.com and @kwgglobal.com) to your email program's "safe senders" list. You will be contacted by both at various points during the production process.

Review comments (if applicable):